# Characterizing non-exponential growth and bimodal cell size distributions in fission yeast: An analytical approach

**Chen Jia[1], Abhyudai Singh[2], Ramon Grima[3]** *

**1** Applied and Computational Mathematics Division, Beijing Computational Science Research Center, Beijing, China, **2** Department of Electrical and Computer Engineering, University of Delaware, Newark, Delaware, United States of America, **3** School of Biological Sciences, University of Edinburgh, Edinburgh, United Kingdom

* Ramon.Grima@ed.ac.uk

**Data Availability Statement:** The MATLAB codes of stochastic simulations of both model I and model II can be found on GitHub via the link

## Abstract

Unlike many single-celled organisms, the growth of fission yeast cells within a cell cycle is not exponential. It is rather characterized by three distinct phases (elongation, septation, and reshaping), each with a different growth rate. Experiments also showed that the distribution of cell size in a lineage can be bimodal, unlike the unimodal distributions measured for the bacterium *Escherichia coli*. Here we construct a detailed stochastic model of cell size dynamics in fission yeast. The theory leads to analytic expressions for the cell size and the birth size distributions, and explains the origin of bimodality seen in experiments. In particular, our theory shows that the left peak in the bimodal distribution is associated with cells in the elongation phase, while the right peak is due to cells in the septation and reshaping phases. We show that the size control strategy, the variability in the added size during a cell cycle, and the fraction of time spent in each of the three cell growth phases have a strong bearing on the shape of the cell size distribution. Furthermore, we infer all the parameters of our model by matching the theoretical cell size and birth size distributions to those from experimental single-cell time-course data for seven different growth conditions. Our method provides a much more accurate means of determining the size control strategy (timer, adder or sizer) than the standard method based on the slope of the best linear fit between the birth and division sizes. We also show that the variability in added size and the strength of size control in fission yeast depend weakly on the temperature but strongly on the culture medium. More importantly, we find that stronger size homeostasis and larger added size variability are required for fission yeast to adapt to unfavorable environmental conditions.

## Author summary

Advances in microscopy enable us to follow single cells over long timescales from which we can understand how their size varies with time and the nature of innate strategies developed to control cell size. These data show that in many cell types, growth is exponential and the distribution of cell size has one peak, namely there is a single characteristic cell

https://github.com/chenjiacsrc/Fission-yeast-cell-size. All data needed to evaluate the conclusions in the paper are present in the paper and in Ref 4.

**Funding:** C.J. acknowledges support from the NSAF grant in National Natural Science Foundation of China with grant No. U1930402. A.S. is supported by the National Institute of Health Grant 1R01GM126557. R.G. acknowledges support from the Leverhulme Trust (RPG-2018-423). The funders played no role in the study design, data collection and analysis, decision to publish, or preparation of the manuscript.

**Competing interests:** The authors have declared that no competing interests exist.

size. However data for fission yeast show remarkable differences: growth is non-exponential and the distribution of cell sizes has two peaks, corresponding to different growth phases. Here we construct a detailed stochastic mathematical model of this organism; by solving the model analytically, we show that it is able to predict the two peaked distributions of cell size seen in data and provide an explanation for each peak in terms of various growth phases of the single-celled organism. Furthermore, by fitting the model to the data, we infer values for the rates of all microscopic processes in our model. This method is shown to provide a much more reliable inference than current methods and shed light on how the strategy used by fission yeast cells to control their size varies with external conditions.

## Introduction

The fission yeast Schizosaccharomyces pombe is a single-cell eukaryote whose shape is well approximated by a cylinder with hemispherical ends [1–3]. The length of the rod-shaped cell increases during the $G_2$ phase of the cell cycle, while its width (diameter) remains almost constant. In experiments, length, area, and volume have all been used to characterize cell size. The recent advent of microfluidic techniques allows the tracking of thousands of individual cells over hundreds of cell cycles which potentially enables a detailed investigation of cell growth and size control strategies [4].

It has been reported that cell size grows exponentially in many cell types such as bacteria, cyanobacteria, archaea, budding yeast, and mammalian cells [5–17]. However, fission yeast undergoes a complex non-exponential growth pattern in each cell cycle, as illustrated by the time-course data of cell size along a typical cell lineage (Fig 1A). At the beginning of the cell cycle, the rod-shaped cell starts to grow by extension at its old cell end (the end that existed before the last division). Later in mid $G_2$ phase, the cell exhibits a transition in cell polarization, and growth is also initiated at the new cell end (the end created during the last division), in a process called new end take-off (NETO) [1, 3]. Cell length increases during the first $\sim$75% of the cell cycle. Cell elongation stops during the remaining $\sim$25% of the cell cycle, when mitosis and cytokinesis occur, and the cell subsequently divides into two almost identical progenies [1, 3].

There has been a long-standing controversy about the growth pattern of fission yeast before mitosis [18–20]. In earlier studies, exponential [19, 21–25], linear [26, 27], and bilinear [1, 3, 28–30] growth models have been proposed. The bilinear model consists of two linear growth regimes with different growth rates separated by a rate change point at $\sim$34% of the cell cycle in wild-type cells, which coincides with NETO [1]. While the predominant viewpoint is that the growth before mitosis is bilinear, more recent data has confirmed exponential growth of mass with some changes in density through the cell cycle [25]. In practice, however, it is very difficult to distinguish the exponential and bilinear growth patterns due to the stochasticity of growth dynamics and the relatively low temporal and spatial resolution of the data.

A remarkable feature of lineage measurements is the bimodal shape of the cell size distribution computed over many cell cycles (Fig 1B). Note that this was not reported in the original paper [4] but stems from an analysis of their published data. Bimodal lineage distributions of fission yeast have not been previously reported in the literature, possibly because such high throughput data have become available only recently. As well, previous studies focused more on the distributions of birth and division sizes [31–33], instead of the distribution of cell size over the whole cell lineage. Actually, the latter contains much more information than the former, since it reflects the full cell cycle dynamics. Recent studies have shown that if cell size

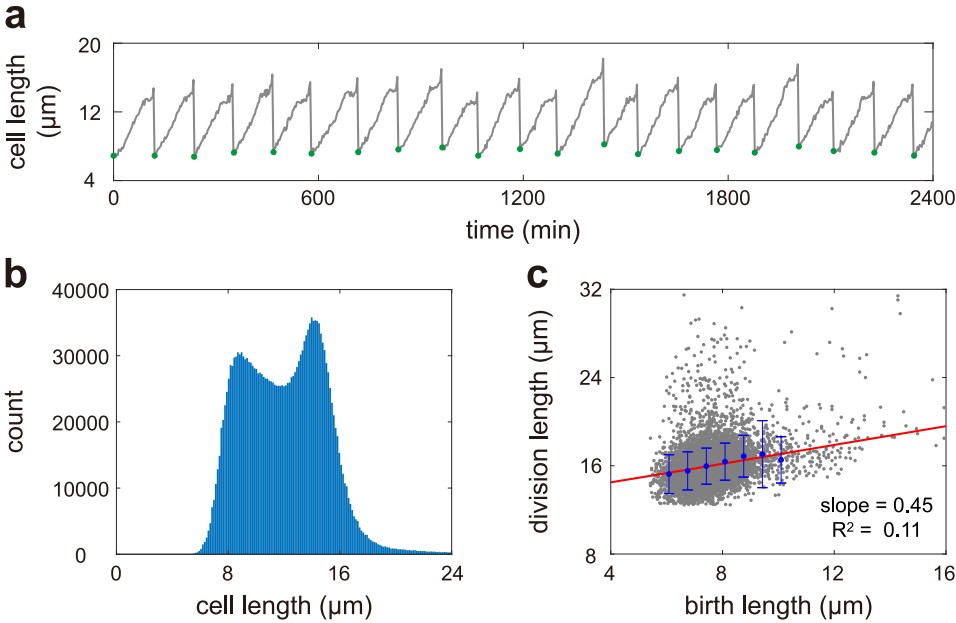

**Fig 1. Cell size dynamics in fission yeast.** **A**: Single-cell time-course data of cell size along a typical cell lineage cultured in the yeast extract medium at 34˚C. Here the size of a cell is characterized by its length. The data shown are published in [4]. The green dots show cell sizes at birth. **B**: Histogram of cell sizes along all cell lineages. The cell size distribution of lineage measurements has a bimodal shape. **C**: Scatter plot of the birth size versus the division size and the associated regression line. When plotting B, C, we use the data of all 1500 cell lineages cultured in the yeast extract medium at 34˚C, each of which is recorded every 3 minutes and is typically composed of 50 − 70 generations. The generation time is 114 ± 23 minutes.

grows exponentially in each generation, then the distribution of cell size must be unimodal [34, 35]. The main aim of the present paper is to propose a detailed model of cell size dynamics in fission yeast that can characterize its non-exponential growth, cell division, and size homeostasis, as well as develop an analytical theory that can account for the bimodal shape of the cell size distribution.

In the study of cell size dynamics, a core issue is to understand the size homeostasis strategies in various cell types, especially in fission yeast [36–51]. There are three popular phenomenological models of cell size control leading to size homeostasis [52]: (i) the timer strategy which implies a constant time between successive divisions regardless of initial size; (ii) the sizer strategy which implies cell division upon attainment of a critical size, and (iii) the adder strategy which implies a constant size addition between consecutive generations. The adder or near-adder behavior has been observed in bacteria, budding yeast, and mammalian cells [10, 11, 16]. However, fission yeast exhibits a sizer-like behavior [36, 53], where cells grow during interphase to a target size of $\sim 14\mu$m in length before entering mitosis and dividing medially, with the standard deviation of the division size being only 7% of the mean [54, 55]. Recently, significant progress has been made to decipher the molecular mechanism responsible for size control in fission yeast and some key proteins have been found to sense cell size and promote mitotic entry [54–58].

A conventional method of inferring the size control strategy is to use the information of cell sizes at birth and at division [36, 59]. This method assumes that the birth size $V_b$ and the division size $V_d$ in each generation are related linearly by

$$V_d = \beta V_b + \gamma + \epsilon, \tag{1}$$

where $0 \leq \beta \leq 2$ and $\gamma \geq 0$ are two constants and $\epsilon$ is a noise term independent of the birth size. Here $\beta$ characterizes the strength of size control with $\beta = 0$, $\beta = 1$, and $\beta = 2$ corresponding to the sizer, adder, and timer strategies, respectively. Using the data of birth and division sizes across generations, the parameter $\beta$ can be determined as the slope of the regression line of the division size on the birth size. The linear relationship with a slope less than 1 between the birth and division sizes in Fig 1C is suggestive of a sizer-like mechanism. However it is also clear that the relationship is very weak with numerous outliers and an exceptionally low $R^2$ around 0.1. This makes the inference of the parameter $\beta$ highly unreliable. Hence another aim of the present paper is to develop a more reliable technique that can be used to accurately infer the size control strategy in fission yeast using a stochastic dynamic approach.

## Results

### Model specification

Here we consider a detailed model of cell size dynamics in fission yeast across many generations, including a complex three-stage growth pattern, asymmetric and stochastic cell division, and size homeostasis (see Fig 2B for an illustration). In our model, cell size can be interpreted as either cell length, area, or volume. Because the cell width is approximately a constant and the cell is rod-like in shape, these three size measures are proportional to each other. The model is based on a number of assumptions that are closely related to lineage data obtained using microfluidic devices. The assumptions are as follows and the specific meaning of all model parameters is listed in Table 1.

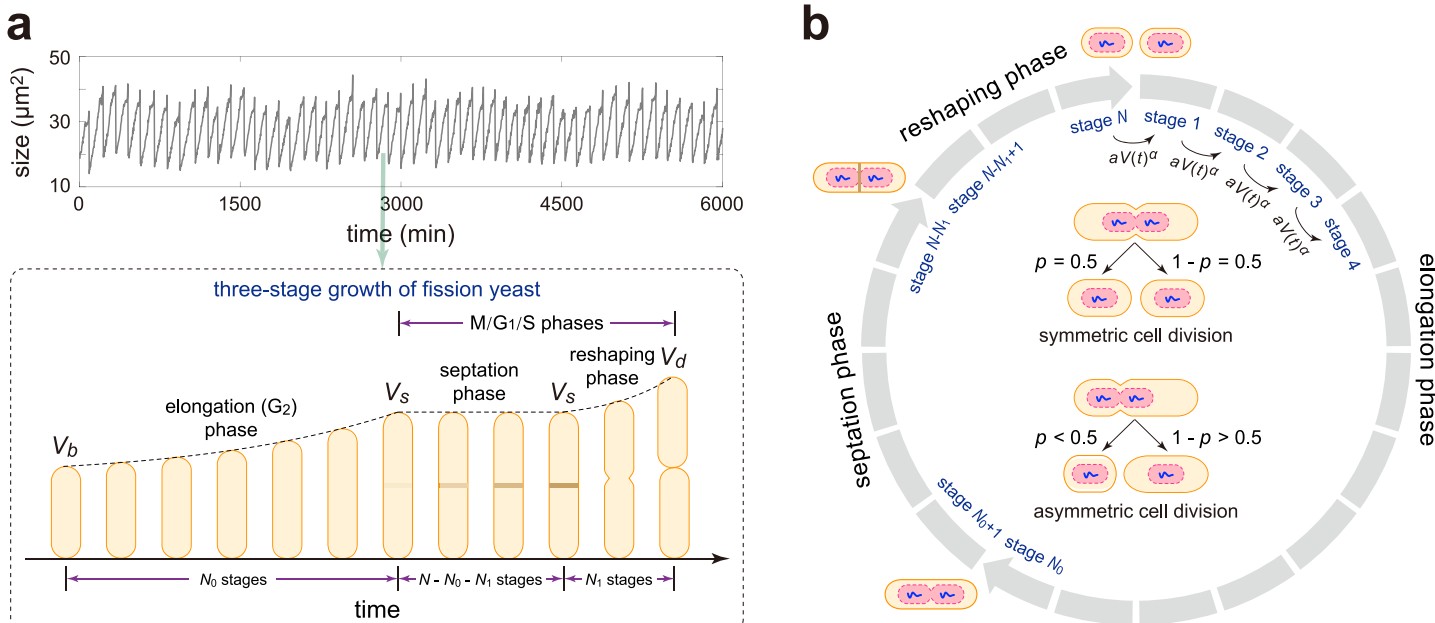

**Fig 2. A detailed model of cell size dynamics in fission yeast. A**: Three-stage growth pattern of fission yeast: an elongation (G$_2$) phase where cell size grows exponentially with rate $g_0$, followed by a septation phase during which the septum is formed and cell size remains constant, and then followed by a reshaping phase where cell length increases abruptly with a higher exponential growth rate $g_1 > g_0$. Here $V_b$ is the size at birth, $V_s$ is the size at septation, and $V_d$ is the size at division. **B**: Schematic illustrating a detailed model of cell size dynamics describing three cell growth phases, size homeostasis, and symmetric or asymmetric partitioning at cell division (see inset graph). Each cell can exist in $N$ effective cell cycle stages. Cell elongation occurs during the first $N_0$ stages, septation occurs during the intermediate $N - N_0 - N_1$ stages, and reshaping of the new cell end occurs during the last $N_1$ stages. The transition rate from one stage to the next at time $t$ is proportional to the $\alpha$th power of the cell size $V(t)$ with $\alpha > 0$ being the strength of size control and $a > 0$ being the proportionality constant. This guarantees that larger cells at birth divide faster than smaller ones to achieve size homeostasis. At stage $N$, a mother cell divides into two daughters that are typically different in size via asymmetric cell division. Symmetric division is the special case where daughters are equisized.

**Table 1. Model parameters and their meaning.**

| model parameters | meaning |
|:---:|:---:|
| $g_0$ | exponential growth rate in the elongation phase |
| $g_1$ | exponential growth rate in the reshaping phase |
| $N$ | total number of effective cell cycle stages |
| $N_0$ | number of cell cycle stages in the elongation phase |
| $N_1$ | number of cell cycle stages in the reshaping phase |
| $r_0$ | proportion of cell cycle stages in the elongation phase |
| $r_1$ | proportion of cell cycle stages in the reshaping phase |
| $w_0$ | proportion of the elongation phase |
| $w_1$ | proportion of the reshaping phase |
| $a$ | proportionality constant for the transition rate between stages |
| $\alpha$ | strength of size control |
| $M_0$ | mean generalized added size in the elongation phase |
| $M_1$ | mean generalized added size in the reshaping phase |
| $p$ | mean partition ratio of cell size at division |

1) The growth of cell size in fission yeast is very different from the exponential growth observed in many other cell types [6]. Actually, fission yeast undergoes a non-exponential three-stage growth pattern: an elongation phase followed by a septation phase and a reshaping phase (Fig 2A) [26]. During the elongation ($G_2$) phase, we assume that the size of each cell grows exponentially with rate $g_0$, which is supported by experiments [19, 21–25]. In some previous papers, the growth before mitosis is assumed to be bilinear with a change in the slope in the mid $G_2$ phase [1, 3, 28–30]. However, bilinear growth is very close to exponential growth and it is very difficult to distinguish them experimentally due to the stochasticity of growth dynamics and the relatively low temporal and spatial resolution of the data.

After the elongation phase, the growth ceases for a period during which the septum is formed [26, 54]. From the lineage data in Fig 1A, it seems also reasonable to assume that there is a small non-zero growth rate in the septation phase. However, according to the principle of parsimony, we choose not to introduce an extra parameter and assume zero growth rate during septation.

At the end of the septation phase, there is a sharp increase in cell length for a short period (a few minutes), during which the new cell end pops out and forms a hemisphere due to turgor pressure [60]. This period is referred to as the reshaping phase since it corresponds to the rounding off of the new end from the septum. Hence intuitively, the mean added size in this phase should roughly equal the size of the two hemispherical end caps. During the reshaping phase, we assume that cell size grows exponentially with a higher rate $g_1 > g_0$. In general, the duration of this phase is very variable and thus this part is often omitted from cell growth studies. Here we choose to include this part into our modelling since it naturally appears in the lineage data (Fig 1A). Since the period of the reshaping phase is very short, the specific growth pattern in this phase does not have much effect on the overall dynamics. The choice of exponential growth during this phase is convenient for an analytical treatment.

2) Each cell can exist in $N$ effective cell cycle stages, denoted by 1, 2, . . ., $N$. Note that the effective cell cycle stages introduced here do not directly correspond to the four biological cell cycle phases ($G_1$, S, $G_2$, and M) or the three growth phases (elongation, septation, and reshaping). Rather, a cell cycle phase or a growth phase corresponds to multiple effective cell cycle stages (Fig 2). Similar assumptions have been successfully used to reproduce the measured variability in cell cycle phase durations in other cell types [61]. We assume that the cell stays in the

elongation phase in the first $N_0$ stages, in the reshaping phase in the last $N_1$ stages, and in the septation phase in the intermediate $N - N_0 - N_1$ stages (Fig 2). The transition rate from one stage to the next at a particular time is proportional to the $\alpha$th power of cell size at that time [35, 62]. In other words, the transition rate between stages at time $t$ is equal to $aV(t)^\alpha$, where $V$($t$) is the cell size at that time, $\alpha > 0$ is the strength of size control, and $a > 0$ is a proportionality constant. Under this assumption, larger cells at birth, on average, have shorter cell cycle duration and lesser volume change than smaller ones; in this way size homeostasis is achieved. Interestingly, under symmetric division and small cell size variability, the size control strength $\alpha$ in our model and the size control strength $\beta$ in the conventional model given in Eq (1) are related by $\beta = 2^{1-\alpha}$ (see Section A in S1 Appendix for the proof).

The entry to mitosis is controlled by a complex gene regulatory network. The cyclin dependent kinase Cdk1, the central mitotic regulator, is regulated by many proteins such as the peripheral membrane kinase Cdr2, the kinase Wee1, the phosphatase Cdc25, and the cyclin B Cdc13 [54]. The prevailing consensus is that the accumulation of regulators upstream of Cdk1, such as Cdr2, Cdc25, and Cdc13, to a critical threshold is required to trigger mitotic entry and cell division, a strategy known as the activator accumulation mechanism [54–58]. Biophysically, the $N$ effective cell cycle stages in our model can be understood as different levels of the key protein that triggers cell division. The power law form for the rate of cell cycle progression may come from cooperation of the key protein, as explained in detail in [35, 62]. This power law not only coincides with certain biophysical mechanisms, but also results in a natural scaling transformation among the timer, adder, and sizer, as will be explained later. Besides, we point out that another strategy called the inhibitor dilution mechanism may also be used for size control. This strategy has been observed in budding yeast [54, 63, 64], where the concentration of Whi5 decreases during the $G_1$ phase due to dilution caused by an increase in cell size. When cell volume is sufficiently large, the Whi5 concentration drops below a given threshold to trigger the $G_1$/S transition allowing subsequent DNA replication and budding.

Let $V_b$ and $V_d$ denote the cell sizes at birth and at division in a particular generation, respectively, and let $V_s$ denote the cell size in the septation phase, which is assumed to be a constant. Then the increment in the $\alpha$th power of cell size, which is referred to as generalized added size, in the elongation phase, $\Delta_0 = V_s^\alpha - V_b^\alpha$, has an Erlang distribution with shape parameter $N_0$ and mean $M_0 = N_0 g_0 \alpha/a$ (see Section A in S1 Appendix for the proof). Similarly, the generalized added size in the reshaping phase, $\Delta_1 = V_d^\alpha - V_s^\alpha$, also has an Erlang distribution with shape parameter $N_1$ and mean $M_1 = N_1 g_1 \alpha/a$. Therefore, the total generalized added size across the cell cycle, $\Delta = \Delta_0 + \Delta_1 = V_d^\alpha - V_b^\alpha$, is the sum of two independent Erlang distributed random variables and has a hypoexponential distribution (also called generalized Erlang distribution) whose Laplace transform is given by

$$\langle e^{-\lambda\Delta} \rangle = \left(1 + \frac{M_0\lambda}{N_0}\right)^{-N_0} \left(1 + \frac{M_1\lambda}{N_1}\right)^{-N_1} := b(\lambda). \qquad (2)$$

Note that $\langle e^{-\lambda\Delta} \rangle \to e^{-(M_0+M_1)\lambda}$ as $N \to \infty$. This means that the generalized added size $\Delta = M_0 + M_1$ becomes deterministic when $N$ is very large. In other words, a higher threshold for the division protein (the key regulator triggering cell division) level results in a less noisy added size and thus a less noisy cell cycle duration. When $N$ is small, the variability in $\Delta$ is much larger. Hence, our model allows the investigation of the influence of added size variability on cell size dynamics. Since the strength of size control and the variability in added size may strongly depend on the culturing condition (strain, medium, temperature) applied, the specific values of $\alpha$ and $N$ may vary a lot in different culturing conditions.

Three special cases deserve special attention. When $\alpha \to 0$, the transition rate between stages is a constant and thus the doubling time has an Erlang distribution that is independent of the birth size; this corresponds to the timer strategy. When $\alpha = 1$, the added size $V_d - V_b$ has an hypoexponential distribution that is independent of the birth size; this corresponds to the adder strategy. When $\alpha \to \infty$, the $\alpha$th power of the division size, $V_d^\alpha$, has a hypoexponential distribution that is independent of the birth size; this corresponds to the sizer strategy. Intermediate strategies are naturally obtained for intermediate values of $\alpha$; timer-like control is obtained when $0 < \alpha < 1$ and sizer-like control is obtained when $1 < \alpha < \infty$ [62].

3) Cell division occurs when the cell transitions from stage $N$ to the next stage 1. At division, most previous papers assume that the mother cell divides into two daughters that are exactly the same in size via symmetric partitioning [31, 65–67]. Experimentally, fission yeast in general do not divide perfectly in half. There has been some evidence indicating that there is a small asymmetry in the position of the septum that is slightly nearer the new end [68, 69]. Here we follow the methodology that we devised in [35, 70] and extend previous models by considering asymmetric partitioning at division: the mother cell divides into two daughters with different sizes.

In this paper, division is assumed to occur after the reshaping phase, i.e. after the rounding off of the two new ends. This is consistent with the lineage data in [4, 26] (see also Fig 1A), as well as Fig 2A in [54]. However, in previous papers [1, 3, 36], division is often assumed to occur after the septation phase—when the septum starts to be degraded, the mother cell has divided into two progenies. Under the latter definition, the division size should be the septation size $V_s$ and the birth size should be the size of a progeny from the septum to the old end. However, since our model is based on the lineage data shown in Fig 1A, where the size of a daughter cell from the septum to the old end is not explicitly given, we choose not to use this definition.

If the partitioning of cell size is symmetric, we track one of the two daughters randomly after division [71, 72]; if the partitioning is asymmetric, we either track the smaller or the larger daughter after division [73, 74]. Let $V_d$ and $V_b'$ denote the cell sizes at division and just after division, respectively. If the partitioning is deterministic, then we have $V_b' = pV_d$, where $0 < p < 1$ is a constant with $p = 0.5$ corresponding to symmetric division, $p < 0.5$ corresponding to smaller daughter tracking, and $p > 0.5$ corresponding to larger daughter tracking. The value of $p$ can be inferred from experiments. However, in fission yeast, the partitioning of cell size is appreciably stochastic. In this case, we assume that the partition ratio $R = V_b'/V_d$ has a beta distribution with mean $p$ [75], whose probability density function is given by

$$h(r) = \frac{1}{B(pv, qv)} r^{pv-1}(1 - r)^{qv-1}, \quad 0 < r < 1, \tag{3}$$

where $B$ is the beta function, $q = 1 - p$, and $v > 0$ is referred to as the sample size parameter. When $v \to \infty$, the variance of the beta distribution tends to zero and thus stochastic partitioning reduces to deterministic partitioning, i.e. $f(r) = \delta(r - p)$.

We next describe our stochastic model of cell size dynamics. The microstate of the cell can be represented by an ordered pair $(k, x)$, where $k$ is the effective cell cycle stage which is a discrete variable and $x$ is the cell size which is a continuous variable. Note that the cell undergoes deterministic growth in each stage (exponential growth in the first $N_0$ and the last $N_1$ stages and no growth in the remaining $N - N_0 - N_1$ stages), and the system can hop between successive stages stochastically. Let $p_k(x)$ denote the probability density function of cell size when the cell is in stage $k$. Then the evolution of cell size dynamics in fission yeast can be described by a

piecewise deterministic Markov process whose master equation is given by

$$\partial_t p_1(x) = -\partial_x[g_0 x p_1(x)] + \int_0^1 \frac{a}{r}\left(\frac{x}{r}\right)^\alpha p_N\left(\frac{x}{r}\right)h(r)dr - ax^\alpha p_1(x),$$

$$\partial_t p_k(x) = -\partial_x[g_0 x p_k(x)] + ax^\alpha p_{k-1}(x) - ax^\alpha p_k(x), \quad 2 \le k \le N_0,$$

$$\partial_t p_k(x) = ax^\alpha p_{k-1}(x) - ax^\alpha p_k(x), \quad N_0 + 1 \le k \le N - N_1,$$

$$\partial_t p_k(x) = -\partial_x[g_1 x p_k(x)] + ax^\alpha p_{k-1}(x) - ax^\alpha p_k(x), \quad N - N_1 + 1 \le k \le N,$$

(4)

where $h(r)$ is the function given in Eq (3). In the first, second, and fourth equations, the first term on the right-hand side describe cell growth and the remaining two terms describe transitions between cell cycle stages. In the third equation, the two terms on the right-hand side describe cell cycle stage transitions. In the first equation, the middle term on the right-hand side describes the partitioning of cell size at division.

## Analytical distribution of cell size for lineage measurements

Let $p(x) = \sum_{k=1}^N p_k(x)$ denote the probability density function of cell size $V$ (here we use $V$ to represent a random variable and use $x$ to represent a realization of $V$). In our model, we assume that the rate of cell cycle progression has a power law dependence on cell size. This assumption implies an important scaling property of our model: if the dynamics for cell size $V$ has a control strength $\alpha$ (with $\alpha < 1$ corresponding to timer-like and $\alpha > 1$ corresponding to sizer-like strategies), then the dynamics for the $\alpha$th power of cell size, $V^\alpha$, has an adder strategy. This scaling property serves as the key to our analytical theory.

Recall that the probability distribution of any random variable with nonnegative values is fully determined by its Laplace transform. To obtain the analytical distribution of cell size along a cell lineage, we introduce $F(\lambda) = \langle e^{-\lambda V^\alpha}\rangle = \int_0^\infty p(x)e^{-\lambda x^\alpha}dx$, which is nothing but the Laplace transform for the $\alpha$th power of cell size. For simplicity, we first focus on the case of deterministic partitioning. Despite the biological complexity described by our model, the Laplace transform can still be solved exactly in steady-state conditions as (see Section B in S1 Appendix for the proof)

$$F(\lambda) = K \int_\lambda^\infty f(u)\prod_{k=0}^\infty b(p^{\alpha k}u)du,$$

(5)

where $b(\lambda)$ is the function given in Eq (2),

$$f(\lambda) = (1 + A_1\lambda)^{N_1}\left[\frac{(1 + A_0\lambda)^{N_0} - 1}{NA_0\lambda} + \frac{N - N_0 - N_1}{N}\right] + \frac{(1 + A_1\lambda)^{N_1} - 1}{NA_1\lambda}$$

(6)

is another function with $A_0 = M_0/N_0$ and $A_1 = M_1/N_1$, and

$$K = \left[\int_0^\infty f(u)\prod_{k=0}^\infty b(p^{\alpha k}u)du\right]^{-1}$$

is a normalization constant. From the definition of $f(\lambda)$ in Eq (6), it is clear that $f(\lambda)$ tends to infinity as $\lambda \to \infty$. However, from the definition of $b(\lambda)$ in Eq (2), the infinite product $\prod_{k=0}^\infty b(p^{\alpha k}\lambda)$ decays to zero as $\lambda \to \infty$ at a faster exponential speed. Hence the integral in Eq (5) is always well defined.

In principle, taking the inverse Laplace transform gives the probability density function of $V^\alpha$, from which the distribution of cell size $V$ can be obtained. Next we introduce how to compute the cell size distribution more effectively using our analytical results. Taking the derivative with respect to $\lambda$ on both sides of Eq (5), using the change of variables formula, and finally replacing $\lambda$ by $i\lambda$ yield (see Section B in S1 Appendix for the proof)

$$\int_0^\infty y\tilde{p}(y)e^{-i\lambda y} = Kf(i\lambda)\prod_{k=0}^\infty b(p^{\alpha k}i\lambda) := G(\lambda), \tag{7}$$

where

$$\tilde{p}(y) = \frac{1}{\alpha}y^{\frac{1}{\alpha}-1}p\left(y^{\frac{1}{\alpha}}\right),$$

is the probability density function of $V^\alpha$. This shows that the Fourier transform of $y\tilde{p}(y)$ is exactly $G(\lambda)$. Since the Fourier transform and inverse Fourier transform are inverses of each other, we only need to take the inverse Fourier transform of $G(\lambda)$ so that we can obtain $y\tilde{p}(y)$. Finally, the cell size distribution $p(x)$ can be recovered from $\tilde{p}(y)$ as

$$p(x) = \alpha x^{\alpha-1}\tilde{p}(x^\alpha). \tag{8}$$

In general, the cell size distribution along a cell lineage can also be numerically computed by carrying out stochastic simulations of the piecewise deterministic Markovian model. However, under the complex three-stage growth pattern of fission yeast, according to our simulations, over $10^7$ stochastic trajectories must be generated in order to obtain an accurate computation of the size distribution (S1 Fig)—this turns out to be very slow. The analytical solution is thus important since it allows a fast exploration of large swathes of parameter space without performing stochastic simulations.

To gain deeper insights into the cell size distribution, we next consider two important special cases. For the case of exponential growth of cell size, there is only the elongation phase and the remaining two phases vanish. In this case, we have $N_1 = 0$ and $N = N_0$; the cell size distribution is still determined by Eq (5) with the functions $b(\lambda)$ and $f(\lambda)$ being simplified greatly as

$$b(\lambda) = (1 + A_0\lambda)^{-N}, \quad f(\lambda) = \frac{(1 + A_0\lambda)^N - 1}{NA_0\lambda}. \tag{9}$$

In fact, the analytical cell size distribution for exponentially growing cells has been studied previously in [35], where the distribution of the logarithmic cell size, instead of the original cell size, is obtained. We emphasize that the analytical expression given here is not only much simpler, but also numerically more accurate than the one given in that paper, which includes the integral of an infinite product term which is difficult to compute accurately.

The second case occurs when $N \to \infty$, while keeping $r_0 = N_0/N$ and $r_1 = N_1/N$ as constant, where $r_0$ and $r_1$ represent the proportions of cell cycle stages in the elongation and reshaping phases, respectively. In this case, the generalized added size $\Delta$ becomes deterministic and the system does not involve any stochasticity. As $N \to \infty$, the Laplace transform given in Eq (5) can be simplified to a large extent as (see Section B in S1 Appendix for the proof)

$$F(\lambda) = K\left\{\frac{r_0}{M_0}\left[E_1(v_b^\alpha\lambda) - E_1(v_m^\alpha\lambda)\right] + \frac{(1 - r_0 - r_1)}{v_m^\alpha}e^{-v_m^\alpha\lambda}\right.$$

$$\left. + \frac{r_1}{M_1}\left[E_1(v_m^\alpha\lambda) - E_1(v_d^\alpha\lambda)\right]\right\}, \tag{10}$$

where $E_1(x) = \int_x^\infty \frac{e^{-u}}{u} du$ is the exponential integral,

$$v_b = p\left(\frac{M_0 + M_1}{1 - p^\alpha}\right)^{\frac{1}{2}}, \quad v_m = \left(\frac{M_0 + M_1 p^\alpha}{1 - p^\alpha}\right)^{\frac{1}{2}}, \quad v_d = \left(\frac{M_0 + M_1}{1 - p^\alpha}\right)^{\frac{1}{2}}$$

are the birth size, septation size, and division size, respectively, and $K = (T_0 + T_s + T_1)^{-1}$ is a normalization constant with

$$T_0 = \frac{\alpha r_0}{M_0}\log\frac{v_m}{v_b}, \quad T_s = \frac{1 - r_0 - r_1}{v_m^\alpha}, \quad T_1 = \frac{\alpha r_1}{M_1}\log\frac{v_d}{v_m}$$

being the durations of the three phases, respectively. Note that in [26], the septation size $V_s$ (size after septation) is called the division size and the division size $V_d$ (size after reshaping) is called the fission size. Here the terminology is slightly different. Taking the inverse Laplace transform finally gives the cell size distribution:

$$p(x) = \frac{w_0}{(\log v_m - \log v_b)x} I_{[v_b, v_m]}(x) + w_s\delta(x - v_m) + \frac{w_1}{(\log v_d - \log v_m)x} I_{[v_m, v_d]}(x), \tag{11}$$

where $I_A(x)$ is the indicator function which takes the value of 1 when $x \in A$ and takes the value of 0 otherwise, $\delta(x)$ is Dirac's delta function, and

$$w_0 = \frac{T_0}{T_0 + T_s + T_1}, \quad w_s = \frac{T_s}{T_0 + T_s + T_1}, \quad w_1 = \frac{T_1}{T_0 + T_s + T_1} \tag{12}$$

are the proportions of subpopulations in the three phases, respectively. This indicates that when added size variability is small, cell size has a distribution that is concentrated on a finite interval between $v_b$ and $v_d$.

To validate our theory, we compare the analytical cell size distribution with the one obtained from stochastic simulations under different choices of $N$ (Fig 3A). Clearly, they coincide perfectly with each other. It can be seen that as added size variability become smaller ($N$ increases), the analytical distribution given in Eq (8) converges to the limit distribution given in Eq (11). When $N$ is small, the size distribution is unimodal. As $N$ increases, the size distribution becomes bimodal with the right peak becoming higher and narrower. The bimodality of the size distribution can be attributed to cells in different phases: the left peak corresponds to cells in the elongation phase and the right peak corresponds to cells in the septation and reshaping phases. When $N$ is very large, the size distribution is the superposition of three terms, corresponding to the three phases of cell growth (see the rightmost panel of Fig 3A).

To gain a deeper insight, we illustrate the cell size distribution as a function of the parameters $\alpha$, $r_0$, $r_1$, and $g_1$ when $N$ is relatively large (Fig 3B–3E). It can be seen that as size control becomes stronger ($\alpha$ increases), the size distribution changes from the unimodal to the bimodal shape (Fig 3B). The size distribution is generally unimodal for timer-like strategies and bimodal for sizer-like strategies. The dependence of the size distribution on $r_0$ is expected —a small $r_0$ results in a small fraction of cells in the elongation phase and thus the left peak is much lower than the right peak, while a large $r_0$ gives rise to the opposite effect (Fig 3C). Bimodality is the most apparent when $r_0$ is neither too large nor too small.

The influence of $r_1$ on the cell size distribution is more complicated. Recall that a larger $r_1$ means a larger fraction of cells in the reshaping phase and a smaller fraction of cells in the septation phase. Here since we fix $r_0$ to be a constant and tune $r_1$, there is little change in the fraction of cells in the elongation phase. As the septation phase becomes shorter ($r_1$ increases), the size distribution changes from being bimodal to being unimodal and becomes more

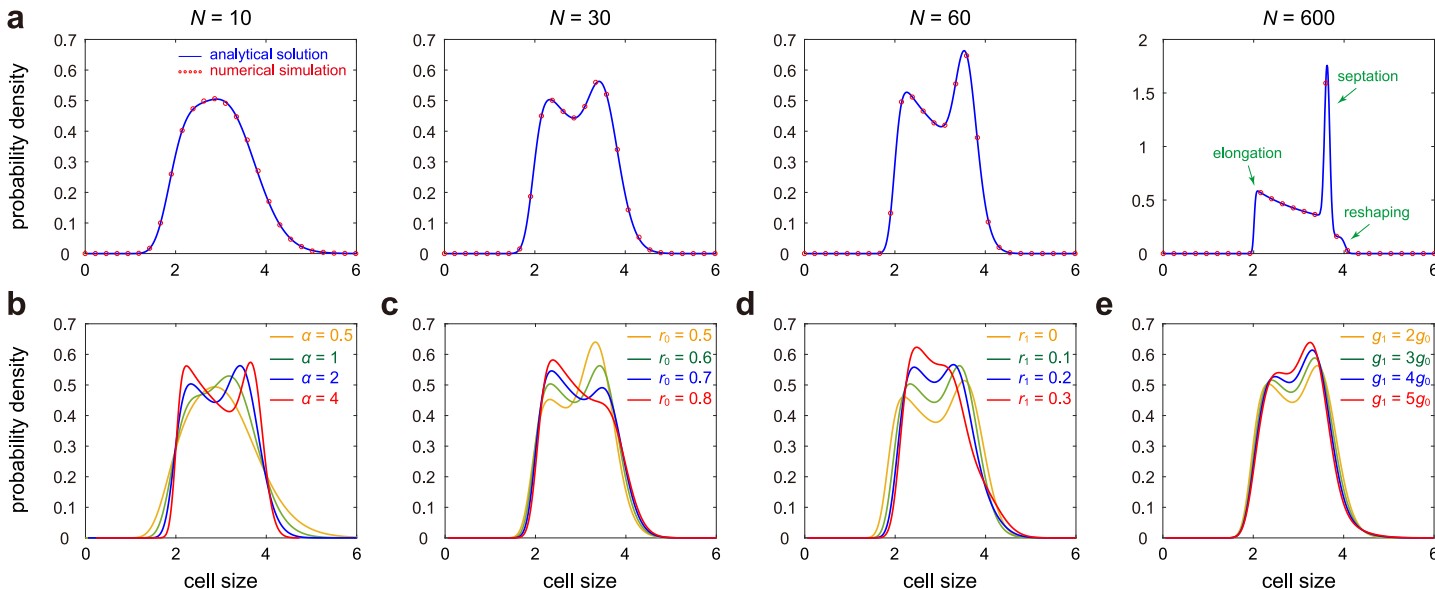

**Fig 3. Influence of model parameters on the cell size distribution. A**: Cell size distribution as $N$ varies. The blue curve shows the analytical distribution obtained by taking the inverse Laplace transform of Eq (5) (e.g. using the technique described by Eqs (7) and (8)) and the red circles show the distribution obtained from stochastic simulations. The parameters are chosen as $r_0 = 0.6$, $r_1 = 0.1$, $g_1 = 2g_0$, $\alpha = 2$. **B**: Cell size distribution as $\alpha$ varies. The parameters are chosen as $N = 30$, $r_0 = 0.6$, $r_1 = 0.1$, $g_1 = 2g_0$. **C**: Cell size distribution as $r_0$ varies. The parameters are chosen as $N = 30$, $r_1 = 0.1$, $g_1 = 2g_0$, $\alpha = 2$. **D**: Cell size distribution as $r_1$ varies. The parameters are chosen as $N = 30$, $r_0 = 0.6$, $g_1 = 2g_0$, $\alpha = 2$. **E**: Cell size distribution as $g_1/g_0$ varies. The parameters are chosen as $N = 30$, $r_0 = 0.6$, $r_1 = 0.1$, $\alpha = 2$. In A-E, the parameters $g_0$ and $p$ are chosen as $g_0 = 0.01$, $p = 0.5$ and the parameters $a$, $M_0$, $M_1$ are chosen so that the mean cell size $\langle V \rangle = 3$.

concentrated (Fig 3D). In particular, bimodality is apparent when the septation phase is relatively long, while a very short septation phase may even destroy bimodality.

Finally, we examine how the cell size distribution depends on the ratio of the growth rate in the reshaping phase to the one in the elongation phase, $g_1/g_0$, which characterizes the sharpness of the size increase in the reshaping phase. As the size addition in the reshaping phase becomes sharper ($g_1/g_0$ increases), the size distribution changes from being bimodal to being unimodal and becomes more concentrated (Fig 3E). Here we fix the mean cell size to be a constant by tuning the parameter $a$ and thus the increase in $g_1$ does not make the right peak shift more to the right. To our surprise, we find that bimodality is the most apparent when the growth rates in the two phases are close to each other, while a very abrupt size addition in the reshaping phase may even destroy bimodality.

To summarize, we find that small added size variability, strong size control, moderate length in the elongation phase, long septation phase, short reshaping phase, and mild size addition in the reshaping phase are capable of producing more apparent bimodality.

## Analytical distribution of the birth size

In our model, the distribution of the birth size $V_b$ can also be derived analytically in steady-state conditions. In fact, the Laplace transform for the $\alpha$th power of the birth size, $V_b^\alpha$, is given by (see Section C in S1 Appendix for the proof)

$$\langle e^{-\lambda V_b^\alpha} \rangle = \prod_{n=1}^{\infty} b(p^{\alpha n} u) = \prod_{n=1}^{\infty} \left( 1 + \frac{M_0 p^{\alpha n} \lambda}{N_0} \right)^{-N_0} \left( 1 + \frac{M_1 p^{\alpha n} \lambda}{N_1} \right)^{-N_1}. \tag{13}$$

Taking the inverse Laplace transform gives the probability density function of $V_b^\alpha$, from which the distribution of $V_b$ can be obtained. A special case takes place when $\alpha$ is large (strong

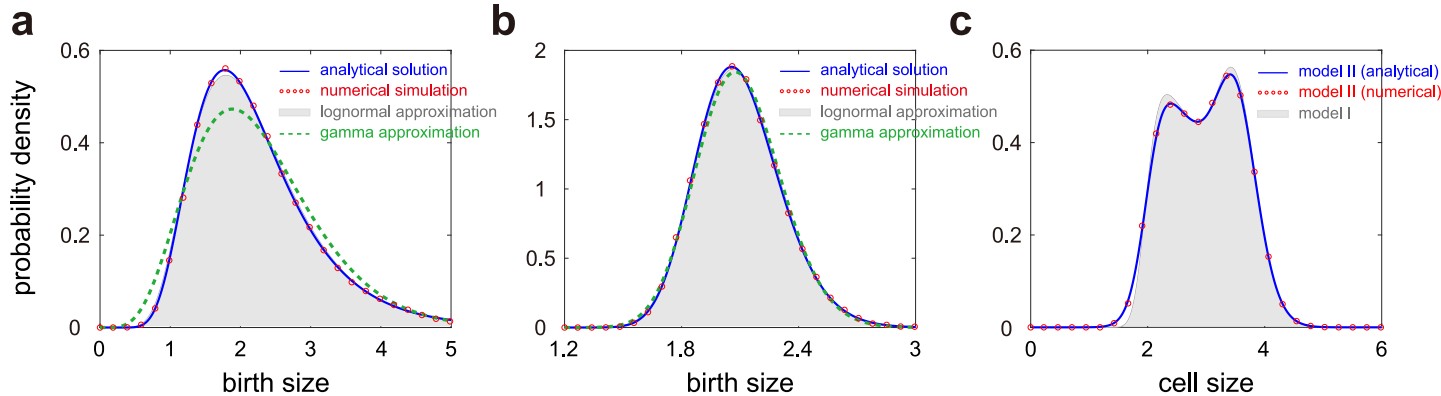

**Fig 4. Further properties of the birth size and cell size distributions. A**: Comparison of the birth size distribution (blue curve and red circles) with its log-normal (solid grey region) and gamma (dashed green curve) approximations when $N$ and $\alpha$ are small. The blue curve shows the analytical distribution obtained by taking the inverse Laplace transform of Eq (13) and the red circles show the distribution obtained from stochastic simulations. **B**: Same as A but when $N$ and $\alpha$ are large. In A, B, the parameters are chosen as $r_0 = 0.6$, $r_1 = 0.1$, $g_0 = 0.01$, $g_1 = 4g_0$, $p = 0.5$. The parameters $N$ and $\alpha$ are chosen as $N = 10$, $\alpha = 0.5$ in A and $N = 30$, $\alpha = 2$ in B. **C**: Comparison between the cell size distributions for the model with deterministic partitioning (solid grey region) and the model with stochastic partitioning (blue curve and red circles). The blue curve shows the analytical distribution obtained by taking the inverse Laplace transform of Eq (15) and the red circles show the simulated distribution. The parameters are chosen as $N = 30$, $r_0 = 0.6$, $r_1 = 0.1$, $g_0 = 0.01$, $g_1 = 4g_0$, $\alpha = 2$, $p = 0.5$. For the model with stochastic partitioning, the parameter $v$ is chosen as $v = 200$. In A-C, the parameters $a$, $M_0$, $M_1$ are chosen so that the mean cell size $\langle V \rangle = 3$ for the model with deterministic partitioning.

size control) or when $p$ is small (smaller daughter tracking). Under the large $\alpha$ or small $p$ approximation, the term $p^{\alpha n}$ is negligible for $n \geq 2$ and it suffices to keep only the first term in the infinite product given in Eq (13). In this case, the laplace transform of $V_b^\alpha$ reduces to

$$\langle e^{-\lambda V_b^\alpha} \rangle = \left(1 + \frac{M_0 p^\alpha \lambda}{N_0}\right)^{-N_0} \left(1 + \frac{M_1 p^\alpha \lambda}{N_1}\right)^{-N_1}.$$

Taking the inverse Laplace transform gives the birth size distribution

$$\mathbb{P}(V_b = x) = \frac{\alpha \beta_0^{N_0} \beta_1^{N_1}}{(N_0 + N_1 - 1)!} x^{\alpha(N_0 + N_1) - 1} e^{-\beta_0 x^\alpha} {}_1F_1(N_1, N_0 + N_1, (\beta_0 - \beta_1)x^\alpha), \qquad (14)$$

where ${}_1F_1$ is the confluent hypergeometric function, $\beta_0 = N_0/M_0\, p^\alpha$, and $\beta_0 = N_1/M_1\, p^\alpha$.

Actually, the birth size distribution has also been computed analytically in some simpler models. It has been shown that the birth size in those models approximately has a log-normal distribution [31] or a gamma distribution [33]. Therefore it is natural to ask whether the birth size in our model shares the same property. To see this, we illustrate the birth size distribution and its approximation by the log-normal and gamma distributions as $N$ and $\alpha$ vary (Fig 4A and 4B). We find that under a wide range of model parameters, the true distribution is in excellent agreement with its log-normal approximation. However, when $N$ and $\alpha$ are both small, the true distribution is severely right-skewed and deviates significantly from its gamma approximation. When $N$ and $\alpha$ are both large, the true distribution becomes more symmetric and the three distributions become almost indistinguishable.

## Influence of stochastic partitioning on the cell size distribution

Thus far, the analytical distribution of cell size is derived when the partitioning at division is deterministic. In the presence of noise in partitioning, we can also obtain an explicit expression for the cell size distribution, whose Laplace transform is given by (see Section B in S1

Appendix for the proof)

$$F(\lambda) = \langle e^{-\lambda V^{\alpha}} \rangle = K \int_{\lambda}^{\infty} f(u) \sum_{n=0}^{\infty} a_n u^n du, \tag{15}$$

where $f(\lambda)$ is the function given in Eq (6),

$$K = \left[ \int_{0}^{\infty} f(u) \sum_{n=0}^{\infty} a_n u^n du \right]^{-1}$$

is a normalization constant, and $a_n$ is a sequence that can be determined by the following recursive relations:

$$a_n = \frac{1}{1 - c_n} \sum_{m=0}^{n-1} a_m c_m b_{n-m}, \quad a_0 = 1. \tag{16}$$

Here $b_n$ and $c_n$ are two other sequences that are defined by

$$b_n = \frac{(-1)^n}{m!(n-m)!} \sum_{m=0}^{n} (N_0)_m (N_1)_{n-m} A_0^m A_1^{n-m}, \quad c_n = \frac{B(\alpha n + pv, qv)}{B(pv, qv)},$$

with $(x)_m = x(x+1)\ldots(x+m-1)$ being the Pochhammer symbol. For the special case of exponential growth of cell size, there is only the elongation phase and the remaining two phases vanish. In this case, we have $N_1 = 0$ and $N = N_0$; the cell size distribution is still determined by Eq (15) with the sequence $b_n$ and the function $f(\lambda)$ being greatly simplified as

$$b_n = \frac{(N)_n (-A_0)^n}{n!}, \quad f(\lambda) = \frac{(1 + A_0\lambda)^N - 1}{NA_0\lambda}.$$

Clearly, fluctuations in partitioning at division lead to a much more complicated analytical expression of the cell size distribution. Actually, when partitioning is stochastic, the size distribution for exponentially growing cells has been derived approximately in [35] under the assumption that noise in partitioning is very small. Here we have removed this assumption and obtained a closed-form solution of the size distribution for general non-exponentially growing cells even if noise in partitioning is very large. Recent cell lineage data suggest that the coefficient of variation of the partition ratio $R = V_b'/V_d$ in fission yeast is 6%—8% under different growth conditions [4].

To see the effect of stochastic partitioning, we illustrate the cell size distributions under deterministic and stochastic partitioning in Fig 4C with the standard deviation of the partition ratio $R$ being 7% of the mean for the latter. Clearly, the analytical solution given in Eq (15) matches the simulation results very well. In addition, it can be seen that noise in partitioning gives rise to larger fluctuations in cell size, characterized by a smaller slope of the left shoulder, an apparent decrease in the height of the left peak, and a slight decrease in the height of the right peak. The valley between the two peaks and the right shoulder are almost the same for the two models.

## Correlation between cell sizes at birth and at division

In [31], it has been shown that the correlation between cell sizes at birth and at division can be used to infer the size control strategy. For the case of deterministic partitioning, since the generalized added size $\Delta = V_d^{\alpha} - V_b^{\alpha}$ is hypoexponentially distributed, it is easy to obtain (see

Section D in S1 Appendix for the proof)

$$\rho(V_b^\alpha, V_d^\alpha) = p^\alpha, \tag{17}$$

where $\rho(X, Y)$ denotes the correlation coefficient between $X$ and $Y$. This characterizes the correlation between birth and division sizes, which only depends on the asymmetry of partitioning ($p$) and the strength of size control ($\alpha$). In particular, we find that if partitioning is deterministic, the correlation is independent of the growth pattern of the cell—both exponentially and non-exponentially growing cells share the same correlation coefficient whenever they have the same $p$ and $\alpha$. Note that in Eq (17), the correlation between $V_b^\alpha$ and $V_d^\alpha$, instead of the correlation between $V_b$ and $V_d$, is computed. This is because of the scaling property of our model: only the generalized added size $V_d^\alpha - V_b^\alpha$ has good analytical properties, instead of the real added size $V_d - V_b$.

In the presence of noise in partitioning, the formula for the correlation coefficient should be modified as (see Section D in S1 Appendix for the proof)

$$\rho(V_b^\alpha, V_d^\alpha) = \sqrt{\frac{\left[(2K_1 + 1)K_2 - K_1^2\right](M_0 + M_1)^2 + K_2\left[\dfrac{M_0^2}{N_0} + \dfrac{M_1^2}{N_1}\right]}{\left[(2K_1 + 1)K_2 - K_1^2\right](M_0 + M_1)^2 + (K_2 + 1)\left[\dfrac{M_0^2}{N_0} + \dfrac{M_1^2}{N_1}\right]}}. \tag{18}$$

where

$$K_1 = \frac{B(\alpha + pv, qv)}{B(pv, qv) - B(\alpha + pv, qv)}, \quad K_2 = \frac{B(2\alpha + pv, qv)}{B(pv, qv) - B(2\alpha + pv, qv)}. \tag{19}$$

In this case, $\rho(V_b^\alpha, V_d^\alpha)$ is generally lower than $p^\alpha$ due to partitioning noise. Interestingly, if partitioning is stochastic, the correlation between birth and division sizes not only depends on $p$ and $\alpha$, but also depends on the parameters $N_0, M_0, N_1, M_1$, which describe the growth pattern of fission yeast. This is very different from the case of deterministic partitioning.

## Experimental validation of the theory

To test our theory, we apply it to the lineage data of cell size in haploid fission yeast that are published in [4]. This data set contains high throughput data of the whole time series of thousands of individual cells over many cell cycles, instead of data at some particular time points [26]. The monitoring of the whole time series allows an accurate inference of all model parameters as well as a deeper understanding of the full cell cycle dynamics.

In this data set, the single-cell time traces of cell area (with unit $\mu m^2$) were recorded every three minutes using microfluidic devices. The experiments were performed under seven growth conditions with different media (Edinburgh minimal medium (EMM) and yeast extract medium (YE)) and different temperatures. For EMM, cells were cultured at four different temperatures (28°C, 30°C, 32°C, and 34°C), while for YE, three different temperatures (28°C, 30°C, and 34°C) were applied. For each growth condition, 1500 cell lineages were tracked and each lineage is typically composed of 50 − 70 generations. Note that for a particular cell lineage, it may occur that the cell was dead or disappeared from the channel during the measurement [4, 76]. Such lineages are removed from the data set and thus the number of lineages used for data analysis for each growth condition is actually less than 1500. In addition, we emphasize that in this data set, the size of a cell is characterized by its area, which has rarely been measured in previous experiments; more commonly used quantities are cell length and cell volume. To make our results more easily comparable to those in the literature, we convert

the cell area data to cell length data (with unit $\mu$m) by using the information of mean cell diameter for each growth condition, which can be estimated from the fluorescence images provided to us by the authors of [4].

Based on the cell length data, it is possible to estimate all the parameters involved in our model for the seven growth conditions. Parameter inference is crucial since it provides insights into the size control strategy, added size variability, and complex growth pattern in fission yeast. We perform parameter inference by fitting the noisy data to two models: the model with deterministic partitioning (model I) and the model with stochastic partitioning (model II). The basic statistics of some important quantities including the birth size $V_b$, the septation size $V_s$, the division size $V_d$, the cell size $V$, the cell diameter $D$, and the cell cycle duration $T$, as well as the estimated values of all parameters for the two models are listed in Table 2. In the following, we briefly describe our parameter estimation method.

1) Estimation of $p$ and $\nu$. Note that the data of cell sizes just before division and just after division, $V_d$ and $V_b'$, across different generations can be easily extracted from the lineage data and thus for model I, the parameter $p$ can be simply estimated as the mean partition ratio

**Table 2. Parameters estimated using lineage data of cell length under seven growth conditions.** The mean and standard deviation (upper-right corner) of six variables are given: the birth size $V_b$, the septation size $V_s$, the division size $V_d$, the cell size $V$, the cell diameter $D$, and the doubling time $T$. We perform parameter inference for both model I and model II. The estimation error for each parameter was computed using bootstrap. Specifically, we performed parameter inference 50 times; for each estimation, the theoretical model was fitted to the data of 50 randomly selected cell lineages. The estimation error was then calculated as the standard deviation over the 50 repeated samplings. The estimates of $p$ and $g_0$ are the same for both models and thus we only list their values for model I.

| statistics | EMM 28°C | EMM 30°C | EMM 32°C | EMM 34°C | YE 28°C | YE 30°C | YE 34°C |
|---|---|---|---|---|---|---|---|
| $V_b$ ($\mu$m) | $7.115^{1.163}$ | $7.214^{1.096}$ | $7.604^{1.329}$ | $7.286^{1.285}$ | $7.066^{1.103}$ | $7.512^{1.036}$ | $7.791^{1.155}$ |
| $V_s$ ($\mu$m) | $13.950^{1.760}$ | $14.386^{1.694}$ | $14.341^{1.689}$ | $14.347^{1.781}$ | $13.480^{1.607}$ | $14.164^{1.649}$ | $14.499^{1.586}$ |
| $V_d$ ($\mu$m) | $15.549^{2.403}$ | $15.748^{2.100}$ | $16.289^{2.812}$ | $15.587^{2.783}$ | $15.189^{2.098}$ | $16.071^{1.971}$ | $16.458^{2.284}$ |
| $V$ ($\mu$m) | $11.159^{2.951}$ | $11.333^{2.960}$ | $11.804^{3.158}$ | $11.364^{3.100}$ | $10.801^{2.931}$ | $11.596^{2.986}$ | $12.078^{3.169}$ |
| $D$ ($\mu$m) | $2.926^{0.282}$ | $2.977^{0.288}$ | $2.958^{0.303}$ | $2.980^{0.292}$ | $2.828^{0.278}$ | $2.981^{0.291}$ | $3.051^{0.318}$ |
| $T$ (h) | $4.121^{1.064}$ | $3.264^{0.667}$ | $3.107^{0.708}$ | $3.633^{1.015}$ | $2.651^{0.477}$ | $2.225^{0.435}$ | $1.906^{0.376}$ |
| model I | EMM 28°C | EMM 30°C | EMM 32°C | EMM 34°C | YE 28°C | YE 30°C | YE 34°C |
| $p$ | $0.459^{0.003}$ | $0.459^{0.002}$ | $0.468^{0.003}$ | $0.470^{0.004}$ | $0.466^{0.003}$ | $0.468^{0.003}$ | $0.475^{0.004}$ |
| $\alpha$ | $1.767^{0.093}$ | $1.695^{0.089}$ | $1.726^{0.102}$ | $1.692^{0.097}$ | $1.139^{0.058}$ | $1.371^{0.072}$ | $1.245^{0.066}$ |
| $N$ | $17.463^{0.803}$ | $20.727^{0.899}$ | $20.002^{0.886}$ | $21.010^{0.900}$ | $32.369^{1.375}$ | $45.713^{1.846}$ | $55.315^{2.126}$ |
| $N_0$ | $11.262^{0.518}$ | $13.646^{0.592}$ | $13.071^{0.579}$ | $14.623^{0.803}$ | $22.950^{0.975}$ | $29.759^{1.202}$ | $35.092^{1.349}$ |
| $N_1$ | $1.214^{0.056}$ | $0.817^{0.035}$ | $0.906^{0.040}$ | $0.714^{0.803}$ | $1.201^{0.051}$ | $1.870^{0.076}$ | $1.864^{0.072}$ |
| $w_0$ | $0.763^{0.015}$ | $0.770^{0.013}$ | $0.765^{0.012}$ | $0.797^{0.017}$ | $0.782^{0.017}$ | $0.742^{0.012}$ | $0.719^{0.011}$ |
| $w_1$ | $0.044^{0.002}$ | $0.025^{0.001}$ | $0.029^{0.001}$ | $0.022^{0.001}$ | $0.027^{0.001}$ | $0.029^{0.001}$ | $0.025^{0.001}$ |
| $a$ | $0.055^{0.003}$ | $0.097^{0.008}$ | $0.085^{0.009}$ | $0.096^{0.008}$ | $0.755^{0.042}$ | $0.682^{0.037}$ | $1.258^{0.071}$ |
| $g_0$ (1/h) | $0.214^{0.023}$ | $0.278^{0.019}$ | $0.280^{0.021}$ | $0.252^{0.029}$ | $0.328^{0.024}$ | $0.396^{0.023}$ | $0.468^{0.027}$ |
| $g_1$ (1/h) | $0.409^{0.044}$ | $0.738^{0.050}$ | $0.767^{0.058}$ | $0.720^{0.083}$ | $0.618^{0.045}$ | $1.240^{0.072}$ | $1.674^{0.097}$ |
| model II | EMM 28°C | EMM 30°C | EMM 32°C | EMM 34°C | YE 28°C | YE 30°C | YE 34°C |
| $\alpha$ | $2.068^{0.152}$ | $1.936^{0.136}$ | $2.068^{0.146}$ | $1.990^{0.139}$ | $1.419^{0.082}$ | $1.622^{0.099}$ | $1.518^{0.090}$ |
| $N$ | $16.387^{0.800}$ | $19.051^{0.925}$ | $18.609^{0.898}$ | $19.499^{0.907}$ | $30.137^{1.382}$ | $43.905^{1.921}$ | $50.067^{2.037}$ |
| $N_0$ | $10.406^{0.508}$ | $12.250^{0.595}$ | $11.984^{0.578}$ | $13.357^{0.621}$ | $21.156^{0.970}$ | $28.099^{1.229}$ | $30.741^{1.251}$ |
| $N_1$ | $1.458^{0.071}$ | $1.124^{0.055}$ | $1.005^{0.049}$ | $0.956^{0.045}$ | $2.170^{0.100}$ | $3.161^{0.138}$ | $3.405^{0.139}$ |
| $w_0$ | $0.770^{0.014}$ | $0.771^{0.015}$ | $0.774^{0.015}$ | $0.803^{0.018}$ | $0.790^{0.016}$ | $0.748^{0.013}$ | $0.718^{0.011}$ |
| $w_1$ | $0.052^{0.002}$ | $0.036^{0.001}$ | $0.032^{0.001}$ | $0.029^{0.001}$ | $0.049^{0.002}$ | $0.048^{0.002}$ | $0.047^{0.002}$ |
| $a$ | $0.025^{0.002}$ | $0.049^{0.003}$ | $0.034^{0.002}$ | $0.043^{0.003}$ | $0.367^{0.018}$ | $0.357^{0.021}$ | $0.574^{0.029}$ |
| $\nu$ | $225.97^{8.68}$ | $257.01^{10.26}$ | $201.98^{7.56}$ | $206.33^{7.84}$ | $198.97^{6.99}$ | $272.09^{11.18}$ | $270.18^{10.85}$ |
| $g_1$ (1/h) | $0.376^{0.040}$ | $0.528^{0.036}$ | $0.713^{0.054}$ | $0.558^{0.064}$ | $0.438^{0.032}$ | $0.796^{0.046}$ | $0.931^{0.054}$ |

$\langle V_b'/V_d \rangle$. For model II, the parameters $p$ and $\nu$ can be inferred by fitting the partition ratio data to a beta distribution.

2) Estimation of $\alpha$. Note that the data of cell sizes at birth and at division, $V_b$ and $V_d$, across different generations can be easily extracted from the lineage data. For model I, since the parameter $p$ has been determined, the strength $\alpha$ of cell size control can be estimated by finding the unique value of $\alpha$ satisfying the equality $\rho(V_b^\alpha, V_d^\alpha) = p^\alpha$. The inference of $\alpha$ for model II is much more complicated. Note that once $\alpha$ is determined, both $K_1$ and $K_2$ can be computed via Eq (19). For model II, the mean and variance for the $\alpha$th power of the birth size are given by (see Section D in S1 Appendix for the proof)

$$\langle V_b^\alpha \rangle = K_1(M_0 + M_1),$$

$$\mathrm{Var}(V_b^\alpha) = \left[(2K_1 + 1)K_2 - K_1^2\right](M_0 + M_1)^2 + K_2\left[\frac{M_0^2}{N_0} + \frac{M_1^2}{N_1}\right].$$

Since $K_1$ and $K_2$ have been determined (assuming $\alpha$ is known), it is possible to estimate both $M_0 + M_1$ and $M_0^2/N_0 + M_1^2/N_1$ using the data of birth sizes. Finally, the control strength $\alpha$ can be estimated by finding the unique value of $\alpha$ satisfying Eq (18).

3) Estimation of $g_0/a$ and $g_1/a$. For model I, the mean and variance for the $\alpha$th power of the birth size are given by (see Section D in S1 Appendix for the proof)

$$\langle V_b^\alpha \rangle = \frac{p^\alpha}{1 - p^\alpha}(M_0 + M_1),$$

$$\mathrm{Var}(V_b^\alpha) = \frac{p^{2\alpha}}{1 - p^{2\alpha}}\left[\frac{M_0^2}{N_0} + \frac{M_1^2}{N_1}\right].$$

Since the parameters $p$ and $\alpha$ have been determined, using the data of birth sizes, we are able to estimate the following two quantities:

$$M_0 + M_1 = N_0\alpha\tilde{g}_0 + N_1\alpha\tilde{g}_1,$$

$$\frac{M_0^2}{N_0} + \frac{M_1^2}{N_1} = N_0\alpha^2\tilde{g}_0^2 + N_1\alpha^2\tilde{g}_1^2,$$

where $\tilde{g}_0 = g_0/a$ and $\tilde{g}_1 = g_1/a$. Once $N_0$ and $N_1$ are known, both $\tilde{g}_0$ and $\tilde{g}_1$ can be solved from the above two equations and thus can be inferred. For model II, we have shown how to estimate $M_0 + M_1$ and $M_0^2/N_0 + M_1^2/N_1$ in step 2).

4) Estimation of $a$, $g_0$, and $g_1$. For each generation, say, the $k$th generation, we fit the time-course data of cell size to a three-stage growth model: an exponential growth in the elongation phase, followed by a constant size in the septation phase and another round of exponential growth in the reshaping phase:

$$V(t) = \begin{cases} V_b e^{g_0(k)t}, & T_k \leq t \leq t_0, \\ V_s, & t_0 \leq t \leq t_1, \\ V_s e^{g_1(k)t}, & t_1 \leq t \leq T_{k+1}, \end{cases}$$

where $T_k$ and $T_{k+1}$ are two successive division times, $g_0(k)$ and $g_1(k)$ are the growth rates in the elongation and reshaping phases for the $k$th generation, respectively, and $t_0$ and $t_1$ are the initial and end times of the septation phase, respectively. By carrying out least-squares optimal fitting, we can estimate the growth rate $g_0(k)$ in the elongation phase and the

growth rate $g_1(k)$ in the reshaping phase for the $k$th generation. Fig 5A illustrates the fitting of the time-course data to the three-stage growth model for three typical cell lineages, from which we can see that the model matches the data reasonably well. Then the parameter $g_0$ can be determined as the mean of $g_0(k)$ across different generations. Since the time that the cell stays in the reshaping phase is very short, the estimate of $g_1(k)$ in general is not accurate. Therefore, we do not adopt this method to estimate the parameter $g_1$. Since both $g_0$ and $\tilde{g}_0 = g_0/a$ have been determined, the parameter $a$ can also be inferred. Since both $a$ and $\tilde{g}_1 = g_1/a$ have been estimated, the parameter $g_1$ can be determined.

5) Estimation of $N$, $N_0$, and $N_1$. Note that once the parameters $N$, $N_0$, and $N_1$ are known, all other parameters can be inferred by carrying out steps 1)—4). Finally, we determine these three parameters by solving the following optimization problem:

$$\min_{N, N_0, N_1} \sum_{i=1}^{M} |p(x_i) - \hat{p}(x_i)|^2, \tag{20}$$

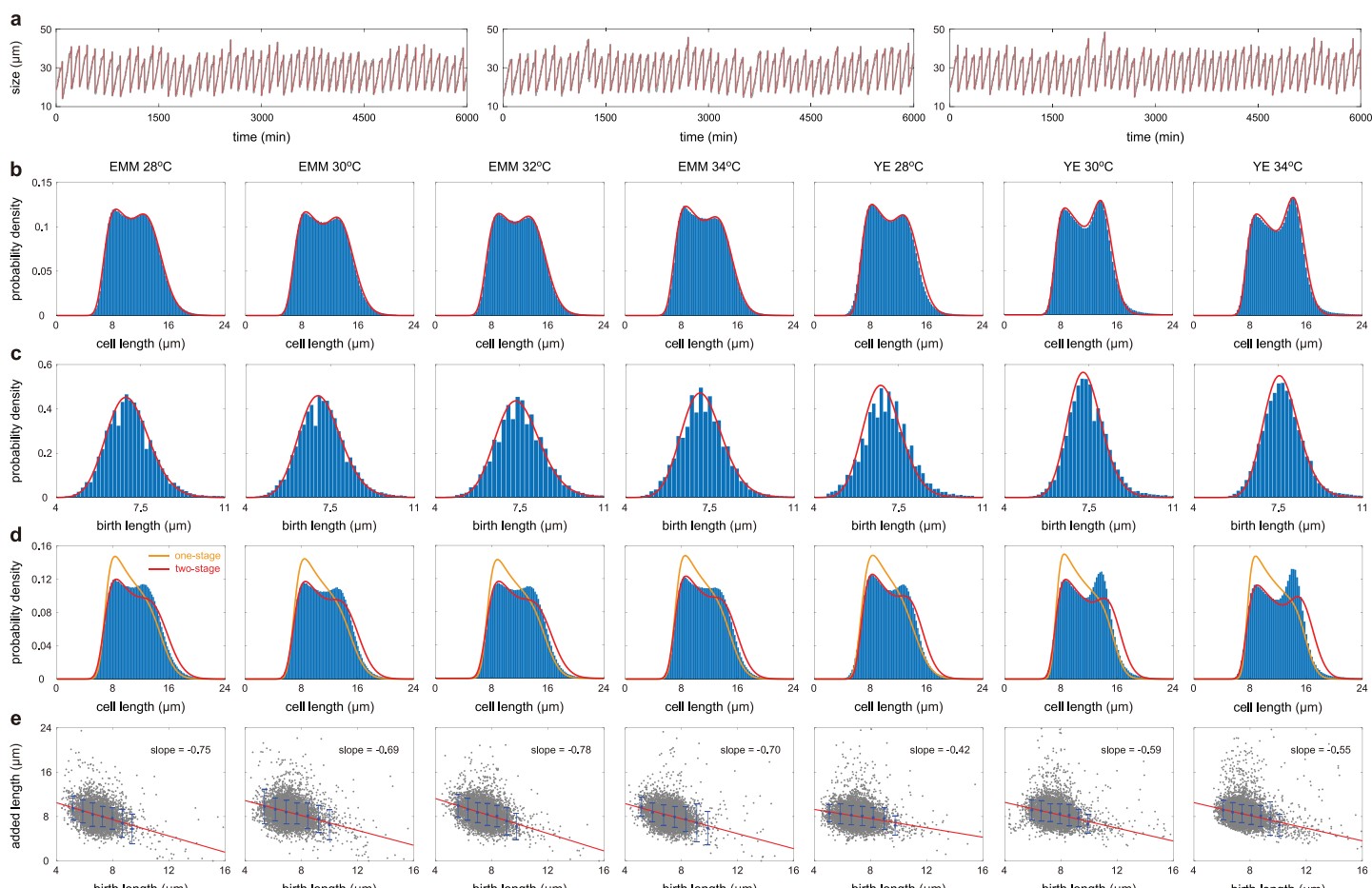

**Fig 5. Fitting experimental data to theory based on the model with deterministic partitioning (model I). A**: Fitting the time-course data of cell size (grey curve) to a three-stage growth model (red curve) for three typical cell lineages cultured in YE at 34˚C. **B**: Experimental cell size distributions (blue bars) and their optimal fitting to model I (red curve) for seven growth conditions. Here the theoretical distributions are computed using Eq (5). **C**: Same as B but for the birth size distributions. Here the theoretical distributions are computed using Eq (13). **D**: Fitting the experimental cell size distributions (blue bars) to the one-stage model with only the elongation phase (orange curve) and the two-stage model with only the elongation and septation phases (red curve). Here the theoretical distributions are computed using Eq (5) by taking $r_0 = 1$ and $r_1 = 0$ for the one-stage model and by taking $r_1 = 0$ for the two-stage model. Both simpler models fail to capture the shape of the cell size distribution. **E**: Scatter plot of the birth length versus the added length and the associated regression line. The slope of the regression line is significantly affected by outliers. To avoid this influence, we removed the outliers identified using Hadi's potential-residual plot [77] and grouped the data into seven bins.

where $p(x)$ is the theoretical cell size distribution, $\hat{p}(x)$ is the sample cell size distribution obtained from lineage data, $x_i$ are some reference points, and $M$ is the number of bins chosen. In other words, we estimate the three parameters by matching the theoretical and experimental cell size distributions. For model I, the theoretical distribution is determined using Eq (5), while for model II, the theoretical distribution is determined using Eq (15). Thus far, all model parameters have been determined.

To test our inference method, we compare the experimental cell size and birth size distributions obtained from the lineage data (blue bars) with the theoretical ones based on the estimated parameters (red curves) under the seven growth conditions for both model I (Fig 5B and 5C) and model II (S2(A) and S2(B) Fig). It can be seen that the cell size distributions of lineage measurements for the seven growth conditions are all bimodal, while the birth size distributions are all unimodal. For the latter model, we also compare the distribution of the partition ratio with its approximation using the beta distribution (S2(C) Fig). Clearly, the theory reproduces the experimental data of fission yeast excellently. Interestingly, while our inference method only involves the matching of the theoretical and experimental cell size distributions, the theoretical birth size distribution also matches the experimental one reasonably well.

To further evaluate the performance of our model, we examine the correlation between the birth size $V_b$ and the division size $V_d$, as well as the correlation between the birth size $V_b$ and the added size $V_d - V_b$. Based on the lineage data, the correlation coefficients for the seven growth conditions are listed in Table 3. The theoretical predictions of the correlation coefficients based on stochastic simulations of model I and model II with the estimated parameters are also listed in Table 3. Clearly, both models capture the birth and division size correlations perfectly. Model I slightly underestimates the birth and added size correlations, while model II slightly overestimates these correlations.

A natural question is whether the lineage data used here can be described by simpler models, such as the one-stage model with only the elongation phase [35, 62] or the two-stage model with only the elongation and septation phases. Here the former only describes the exponential growth in the G$_2$ phase, while the latter ignores the abrupt increase in cell length due to the rounding off of the new ends. To see this, we also fit the cell size distribution to the one-stage and two-stage models by using the inference method introduced above (Fig 5D). Both simpler models fail to capture the unusual shape of the cell size distribution. The one-stage model always predicts a unimodal distribution; the two-stage model predicts a unimodal distribution for EMM and a bimodal distribution for YE. While the two-stage model excellently captures the left peak, it fails to reproduce the right peak due to neglection of the reshaping phase. This suggests that our model is the simplest model that can describe the lineage data of fission yeast.

**Table 3. Correlation coefficients between the birth and division (added) sizes for the seven growth conditions.** The experimental correlation coefficients are computed using the lineage data, while the theoretical correlation coefficients are computed using stochastic simulations based on model I and model II.

| $\rho(V_b, V_d)$ | EMM 28˚C | EMM 30˚C | EMM 32˚C | EMM 34˚C | YE 28˚C | YE 30˚C | YE 34˚C |
|---|---|---|---|---|---|---|---|
| experiment | 0.2599 | 0.2834 | 0.2885 | 0.2753 | 0.4232 | 0.3534 | 0.3999 |
| model I | 0.2576 | 0.2704 | 0.2734 | 0.2844 | 0.4201 | 0.3544 | 0.3959 |
| model II | 0.2432 | 0.2604 | 0.2573 | 0.2740 | 0.4182 | 0.3669 | 0.4051 |
| $\rho(V_b, V_d - V_b)$ | EMM 28˚C | EMM 30˚C | EMM 32˚C | EMM 34˚C | YE 28˚C | YE 30˚C | YE 34˚C |
| experiment | -0.2261 | -0.2415 | -0.1888 | -0.2431 | -0.1124 | -0.1812 | -0.1143 |
| model I | -0.2063 | -0.1924 | -0.1966 | -0.1927 | -0.0511 | -0.1233 | -0.0853 |
| model II | -0.2851 | -0.2678 | -0.2990 | -0.2900 | -0.1683 | -0.2269 | -0.2000 |

The perfect match between experiments and the three-stage model and the poor match between experiments and simpler models support the main assumptions of the three-stage growth model and the choice of the rate of moving from one cell cycle stage to the next to be a power law of cell size.

The parameters estimated in Table 2 also provide some useful insights of biological interest. Compared with model II, model I gives rise to a lower estimate of $\alpha$ and $r_1$, as well as a higher estimate of $N$, $a$, and $g_1$; the estimates of other parameters are very similar for the two models. While both model I and model II capture lineage data very well, we next base our discussion on the parameters of model II since this is biologically more realistic and thus its parameter estimates are more reliable.

First, our data analysis reveals some significant differences between the two media applied. From Table 2, it can be seen that cells cultured in EMM have a relatively strong size control (large $\alpha$) and a relatively large added size variability (small $N$), while cells cultured in YE have a relatively weak size control (small $\alpha$) and a relatively small added size variability (large $N$). Furthermore, we find that the size control strategy in fission yeast is sizer-like for all the seven growth conditions: for model II, the strength $\alpha$ of size control is typically 2.0 for EMM and is typically 1.5 for YE. This is in sharp contrast to the adder strategy found in *E. coli*, where $\alpha$ is estimated to be $0.8 - 1.2$ for different growth conditions [35]. Our result confirms the previous finding that fission yeast uses a sizer-like strategy to achieve size homeostasis [36, 53]. The sizer-like strategy also agrees with our theoretical result that bimodal size distributions are more likely to occur in a sizer than in a timer. Based on the lineage data, the mean septation length $\langle V_s \rangle$, i.e. the mean cell length before mitotic entry, is estimated to be $13.5 - 14.5 \, \mu$m for all growth conditions, which is very stable. This is consistent with the experimental value measured in many previous papers [28, 36, 42, 78, 79]. Recall that the septation length defined here is the division size defined in those papers since the reshaping phase is assumed to belong to the previous cell cycle in the present paper.

Based on the estimated parameters for the seven growth conditions, we observed a strong linear relationship between $g_0$ and $N$, as well as a weaker linear relationship between $g_0$ and $\alpha$ (Fig 6A and 6B). A higher growth rate in the elongation phase gives rise to a larger number of cell cycle stages and a smaller strength of size control. The positive correlation between $g_0$ and $N$ implies that under unfavorable environmental conditions, a lower threshold level of the division protein is used to promote mitotic entry and trigger cell division, which causes larger added size variability. The negative correlation between $g_0$ and $\alpha$ suggests that a stronger size control is required for fission yeast to adapt to unfavorable conditions.

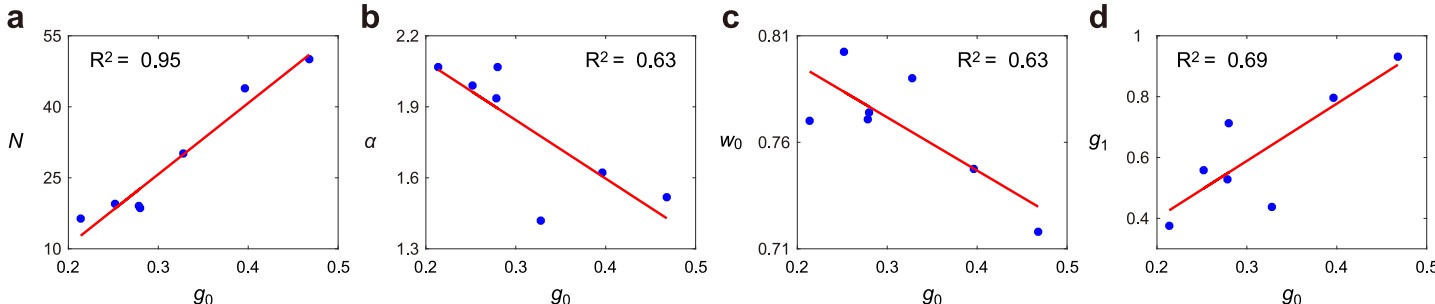

**Fig 6. Connection between the estimated model parameters of model II. A:** Scatter plot of the estimated $N$ versus the estimated $g_0$ under seven growth conditions. **B:** Scatter plot of the estimated $\alpha$ versus the estimated $g_0$. **C:** Scatter plot of the estimated $w_0$ versus the estimated $g_0$. **D:** Scatter plot of the estimated $g_1$ versus the estimated $g_0$. The red lines shown in A-D are regression lines.

To further validate our theory, we illustrate the scatter plots of the birth length versus the added length and the corresponding regression lines in Fig 5E for all growth conditions. A slope of −1 for this regression line is indicative of the sizer strategy, while a slope of 0 implies the adder strategy. Such plots have been reported many times in the literature for wild-type haploid cells and generally show a clear negative correlation with a slope between −0.7 and −0.9 [28, 36, 51, 53], while there is also a study showing that the slope can be as low as −0.37 for wild-type diploids [28]. Based on the lineage data, we find that the slope is typically −0.7 for EMM, which is consistent with the values reported in previous studies, while the slope is typically −0.5 for YE. The reason why YE has a smaller slope is probably due to the fact that cells cultured in YE have faster growth rates, which give rise to weaker size control (Fig 6B). Note that similar phenomenon has also been observed in *E. coli*, where an adder-like behavior was found in fast growth conditions, while a sizer-like behavior was found at low growth rates [80].

In addition, our data analysis also provides the estimation of the fractions of cell cycle in the three phases. Note that the fraction of the elongation (reshaping) phase is not simply given by $r_0$ ($r_1$). This is because the transition rate between cell cycle stages is an increasing function of cell size, which means that earlier (later) stages have longer (shorter) durations. The real fraction of each growth phase can be estimated approximately via Eq (12). According to our estimation, the proportion $w_0$ of the elongation phase is about $72\% - 80\%$ of the cell cycle and the proportion $w_1$ of the reshaping phase is only $3\% - 5\%$ for all growth conditions (Table 2). This is consistent with the previous result that cells elongate during the first $\sim 75\%$ of the cell cycle [1]. In addition, the mean duration $w_1 \langle T \rangle$ in the reshaping phase, i.e. the time needed for cells to form hemispherical new ends from the septum, is estimated to be $5 - 8$ min for all growth conditions (except EMM at 28˚C) and the mean length increase $\langle V_d \rangle - \langle V_s \rangle$ in the reshaping phase, i.e. the total length of the two hemispherical end caps, is estimated to be $1.2 - 2$ $\mu$m for all growth conditions. This coincides with the previous observation that the length of each daughter cell grows by $\sim 1$ $\mu$m within 5 min after septation [3].

Interestingly, from the estimated parameters, we also observed a negative correlation between $g_0$ and $w_0$ and a positive correlation between $g_0$ and $g_1$ (Fig 6C and 6D). This means that a higher growth rate in the elongation phase is associated with a smaller proportion of the elongation phase and a higher growth rate in the reshaping phase. The negative correlation between $g_0$ and $w_0$ can be explained as follows: a higher growth rate in the $G_2$ phase gives rises to a shorter time to reach the target length of $\sim 14$ $\mu$m before mitosis and thus results in the smaller proportion of that phase. In addition, the slope of Fig 6D is estimated to be 1.88, implying that the growth rate of cell length in the rephrasing phase is about twice as large as that in the elongation phase.

An interesting feature implied by the fission yeast data is that the mean partition ratio $p$ is $0.46 - 0.48$ for all growth conditions, implying that cell division is possibly asymmetric (Table 2). Although $p$ only slightly deviates from 0.5, this difference is significant with a $p$-value less than 0.001 for each growth condition according to the sensitive $Z$-test. This deviation may result from the asymmetry in the position of the septum which is slightly nearer the new end [68, 69]. However, we cannot exclude the possibility that the partitioning is actually symmetric and the deviation of $p$ from 0.5 is an artifact due to the segmentation algorithm used in [4], where the old-pole tips tend to be cut (S3 Fig).

## Conclusions and discussion

In this work, we proposed two detailed models of cell size dynamics in fission yeast across many generations and analytically derived the cell size and birth size distributions of measurements obtained from a cell lineage. The main feature of cell size dynamics in fission yeast is its

three-stage non-exponential growth pattern: a slow growth in the elongation phase, an arrest of growth in the septation phase, and a rapid elongation in the reshaping phase. The first model assumes that (i) the cell undergoes deterministic exponential growth in the elongation and reshaping phases with the growth rate in the latter phase being greater than that in the former phase; (ii) the size remains constant in the septation phase; (iii) the size just after division is a fixed fraction of the one just before division; (iv) the cell cycle is divided into multiple effective cell cycle stages which correspond to different levels of the division protein which triggers cell division; (v) the rate of moving from one stage to the next has a power law dependence on cell size. A second model was also solved which relaxes assumption (iii) by allowing the size just after division to be a stochastic fraction of the one just before division with the fraction being sampled according to a beta distribution. Under assumptions (iv) and (v), the three typical strategies of size homeostasis (timer, adder, and sizer) are unified.

Experimentally, the cell size distribution of lineage data in fission yeast is typically bimodal under various growth conditions. This is very different from the unimodal size distribution observed in many other cell types [35]. Interestingly, the bimodal cell size distribution of fission yeast can be excellently reproduced by the analytical solutions of both models. The origin of bimodality is further investigated and clarified in detail; we find that bimodality becomes apparent when (i) the variability in added size is not too large, (ii) the strength of size control is not too weak, which implies that adder or sizer-like strategies enforce size homeostasis, (iii) the proportion of the elongation phase is neither too large nor too small, (iv) the proportion of the septation phase is large, (v) the proportion of the reshaping phase is small, and (vi) the size addition in the reshaping phase is not too sharp. We also find that fluctuations in partitioning at division has a considerable influence on the shape of the cell size distribution by declining the slope of the left shoulder, as well as lowering the heights of the two peaks.

Furthermore, we have developed an effective method of inferring all the parameters involved in both models using single-cell lineage measurements of fission yeast based on the information of (i) the partition ratio, namely, the ratio of the size just after division to the size just before division, across different generations, (ii) the mean and variance of the birth size across different generations, (iii) the correlation of cell sizes at birth and at division, and (iv) the cell size distribution. Specifically, we infer the parameters except the numbers of cell cycle stages in different phases using the information (i)-(iii) and then determine the remaining parameters by matching the theoretical and experimental cell size distributions.

We have shown that the theoretical cell size and birth size distributions provide an excellent fit to the experimental ones of fission yeast reported in [4] under seven different growth conditions. This match provides support for two implicit important assumptions of our model: (i) the cell undergoes a complex three-stage growth pattern and (ii) the speed of the cell cycle progression (the transition rate between cell cycle stages) depends on cell size in a power law form. Finally, based on matching the experimental to the theoretical cell size distributions, we have estimated all model parameters from lineage data. Simulations with the inferred parameters using distribution matching also captured the correlation between birth and division sizes, and between birth and added sizes—this provides further evidence of the accuracy of our detailed model.

Based on the estimated parameters, we confirmed the previous result that fission yeast has a sizer-like control strategy. Cells cultured in EMM have a larger added size variability and a stronger size control than those cultured in YE. The estimated values of the mean septation length, the proportion of the elongation phase, as well as the duration and length increase in the reshaping phase are all consistent with the literature. Moreover, we also observed a negative correlation between the growth rate and (i) the added size variability, (ii) the size control strength, and (iii) the proportion of the elongation phase. This reveals that stronger size

homeostasis and larger cell cycle duration variability are required in slow growth conditions. The growth rate in the reshaping phase was found to be twice that in the elongation phase.

Further research aims to develop more realistic models which coordinate cell size dynamics with gene expression dynamics in fission yeast and investigate the corresponding concentration homeostasis of mRNAs and proteins [81].

## Methods

All methods can be found in the main text and in S1 Appendix.

## Supporting information

**S1 Appendix. Mathematical details.** This file contains the mathematical details of the stochastic cell size model, as well as the detailed derivations of the cell size distribution, the birth size distribution, and the correlation between birth and division sizes.
(PDF)

**S1 Fig. Cell size distributions obtained using stochastic simulations.** The three distributions are obtained by generating $10^5$, $10^6$, and $10^7$ stochastic trajectories, respectively. The parameters are chosen as $N = 50$, $r_0 = 0.6$, $r_1 = 0.1$, $g_0 = 0.01$, $g_1 = 2g_0$, $\alpha = 2$, $p = 0.5$. The parameters $a$, $M_0$, $M_1$ are chosen so that the mean cell size $\langle V \rangle = 3$.
(EPS)

**S2 Fig. Fitting experimental data to theory based on the model with stochastic partitioning (model II). A**: Experimental cell size distributions (blue bars) and their optimal fitting to model II (red curve) for seven growth conditions. Here the theoretical distributions are computed using Eq (15). **B**: Same as A but for the birth size distributions. Here the theoretical distributions are computed using stochastic simulations. **C**: Same as A but for the partition ratio distributions. Here the theoretical distributions are computed using Eq (3).
(EPS)

**S3 Fig. Fluorescence image of fission yeast cells and the segmentation algorithm used in [4] to identify the outline of a new born cell.** At division, the segmentation algorithm tends to cut old-pole tips.
(TIFF)

## Acknowledgments

We are grateful to Professor Nakoaka and Professor Wakamoto for sending us the fluorescence images of fission yeast that we used to estimate the cell diameter.

## Author Contributions

**Conceptualization:** Ramon Grima.

**Formal analysis:** Chen Jia.

**Investigation:** Chen Jia.

**Methodology:** Chen Jia.

**Supervision:** Abhyudai Singh, Ramon Grima.

**Writing – original draft:** Chen Jia, Ramon Grima.

**Writing – review & editing:** Chen Jia, Abhyudai Singh, Ramon Grima.

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
