## [Decision Letter · Decision Letter 0]

10 Aug 2021

Dear Prof. Grima,

Thank you very much for submitting your manuscript "Characterizing non-exponential growth and bimodal cell size distributions in Schizosaccharomyces pombe: an analytical approach" for consideration at PLOS Computational Biology.

As with all papers reviewed by the journal, your manuscript was reviewed by members of the editorial board and by several independent reviewers. In light of the reviews (below this email), we would like to invite the resubmission of a significantly-revised version that takes into account the reviewers' comments.

We cannot make any decision about publication until we have seen the revised manuscript and your response to the reviewers' comments. Your revised manuscript is also likely to be sent to reviewers for further evaluation.

Sincerely,

Attila Csikász-Nagy

Associate Editor

PLOS Computational Biology

Jason Haugh

Deputy Editor

PLOS Computational Biology

Reviewer's Responses to Questions

**Comments to the Authors:**

Reviewer #1: I am not and expert in the computational approaches employed in this manuscript so I have restricted my comments to those that related to the biological context and significance of the authors results.

Major points:

(1) The introduction fails to adequately summarise what is known about how fission yeast control their size given that there is there is an extensive amount of literature that the authors have failed to properly introduce. A more detailed account of existing literature is required given the subject matter of the article. In a similar vein it is kind of inexplicable that the authors do not even cite Fantes ’77 given it is the foundational study for growth and size control of fission yeast.

Similarly, there is also a significant amount of work that describes the growth patterns of fission yeast as a combination of multiple linear (or one exponential, depending on the study) growth phases with a constant length period during mitosis and separation before division. Again, this is not really discussed at all in the introduction and the foundational studies including original account of these phenomena Mitchinson and Nurse ’85 are not even cite, let alone discussed despite being highly relevant.

(2) the authors state that the growth in “the elongation phase is assumed to be linear.” This is misleading to the point of being wrong. There is very little evidence that the growth phase in fission yeast is a simple linear growth. The predominant idea is that elongation phase is bi-linear, i.e. it is constituted of two linear growth regimes with different growth rates separated by a rate change point. In reality this is very close to an exponential growth pattern and it is very difficult to distinguish these two possibilities experimentally.

(3) Regarding the authors statement that “the linear relationship between birth and division sizes are actually very weak” which “makes the inference of the parameter β highly unreliable”. Sizer and adder models are more typically assessed using the relationship between birth size and ∆volume. These relationships have been published many times since originally described by Fantes in 1977 and have generally shown very robust correlations. This relationship is critical and must be shown for both the data and the models including the in an updated version of table 3.

(4) The authors assume that there is a short increase in cell size prior to division and model this as an exponential growth rate with a higher exponent than that in the subsequent elongation phase. When a cell divides the new end forms a semi-sphere which pops out due to hydrostatic pressure.

Is it not possible / likely that this increase in size in the fission phase the authors are modeling is due to expansion of the new ends due to cell division and hydrostatic pressure? If so there is no basis to the assumption this volume growth is exponential because the semi-sphere that pops out is presumably always approximately the same size and takes the same time. This needs to be investigated carefully and the model updated accordingly.

(5) Throughout the authors refer to cell size which is generally term that encompasses many cellular parameters. In reality the authors are only looking at cell area, rather than dry mass, buoyant mass, total protein content etc. The text should be updated accordingly. Moreover, it would be far more useful if the authors converted their area estimates to volume for more ready comparison between studies.

Reviewer #2: The manuscript of Jia et al. contains an original modelling approach for cell size distributions in fission yeast cultures. It is clearly written, the applied methodology is correct and innovative, the drawn conlusions are important. The computational simulations are well fitted to previously published experimental data, therefore the results of this paper may give useful thoughts to both modelers and experimentalists of the yeast cell cycle field in the future. Size homeostasis is a hot topic in today's cell physiological studies, which justifies the significance of this research. I recommend the publication of this manuscript is PLOS Comp Biol, however, there are several points (listed below) needed to be clarified and fixed first, therefore a major revision is required to my mind.

1.) The reference list contains rather few items and their selection seems to be strange. Please note that there is a huge number of published papers from the last 30 years concerning cellular growth and size control in fission yeast as well as their molecular background (I upload a list, which is also far from full). By contrast, among the authors 33 items, only 7 deals very strictly with the experimental background of this theoretical work. In the other papers I attach, there are lots of data for size distributions, growing patterns, size control strength in fission yeast cultures, moreover, preoteins and genes involved in these phenomena, etc. The authors should refer to a much broader list and also explain why they chose exclusively the data of Nakaoka and Wakamoto (PLOS Biol, 2017) for their model fitting.

2.) A problem (the smaller one) with the Nakaoka-Wakamoto paper (and evidently with the simulations) is that cell size is given as "area" in dimension of micrometer^2. This is difficult to measure, rarely used, and therefore the numbers practically do not say anything to specialists of this field. Instead, mainly "cell length" (in microns) or perhaps "cell volume" (in femtoliters) are the commonly used parameters for fission yeast cell size, which are generally given in experimental papers, therefore simulations of these parameters could be easily compared to literature data.

3.) The much larger problem with the simulations and the experimental data behind is the following. What the authors call "fission phase" (explained in their Fig. 2), is generally thought to belong to the next cell cycle, i.e., what they call Vs is generally called Vd, and Vb is also generally thought to be quite different. When the primary septum starts to become degraded, then we consider that the mother cell has divided into two progenies. Defining the birth and division times and sizes the way given in this manuscript once seems to be difficult to be defined precisely. Moreover, the authors study a size range different from the conventionally used one. Finally, I have a feeling that even the bimodal size distribution might be an artefact of the incorrectly positioned division times. To my mind, it is not a general view that size distribution in S. pombe cultures were bimodal, even I cannot fine the bimodal experimental histograms in the Nakaoka-Wakamoto paper. So, I cannot accept the sentence from the Author summary saying that "two characteristic cell sizes exist". This is probably the main point needed to be either discussed in detail or fixed in a revised version!

4.) There is also another bombast, but incorrect sentence in the Author summary saying that "we construct the first mathematical model of this organism".

5.) To my mind, whether "beta = 2 corresponds to the timer strategy" (page 3) depends on the growth mode; it is correct in case of a pure exponential growth pattern only. Moreover, this parameter beta is often called the strength of size control in the literature. By contrast, the author define an alpha parameter (pages 5, 6), which is called the strength of size control. Will the authors give us the connection between these alpha and beta parameters?

6.) In the legend to Fig. 1, the phrase "length of each generation" should be replaced by "generation time".

7.) Please note that even if a fission yeast cell divides asymmetrically, the diameters of the progenies are usually equal, therefore there is a mistake in Fig. 2b.

8.) The authors suppose that growth is exponential in the elongation phase (characterized by a g0 parameter), although they mention that "in some previous papers it is assumed to be linear" (page 5). By contrast, this debate has not been resolved by now, and bilinear growth was also found in a paper in 2021. This should be discussed in the paper.

9.) The authors suppose that growth is also exponential in the fission phase (characterized by a g1 > g0 parameter). Besides my point 2.), there is another problem here. It is generally assumed that in this stage cell elongation is not really growth, but it is rather the rounding off of the new cell ends from the septum, which is driven by mechanical forces mainly. The duration of this phase is very variable and it probably depends on geometrical and osmotic factors. The way how cells elongate here shows abnormalities, therefore this part is often omitted from cell length growth studies. The authors should explain their exponential hypothesis.

10.) The model supposes that the cell cycle consists of effective stages (N, N0, N1) and transitions from one to the next, but this is obscure. Although some cell cycle transitions are cytologically known, like the G1/S, G2/M and the metaphase/anaphase transitions, the authors should give us clear ideas on what they are talking about. For example, they mention some mysterious "division proteins", but their examples (Cdc13, Cdc25, Cdr2) are probably all required for the same G2/M transition. Please also note that these proteins are often regulated post-translationally, therefore their activities matter rather than their levels (page 6). The value of these N parameters is also interesting. In Table 2, it is given somewhere between 16 and 55, which seems to be unexpectedly large. Moreover, if it characterizes the number of cell cycle stages, how might it depend on the culturing techniques (medium, temperature) applied?

11.) I think that r1 = N1/N, so there is a wrong lowercase "n" in page 9.

12.) In Fig. 3. it should be more clearly indicated which parameter set gives the best fit to Fig. 1.

13.) The correlation coefficient between Vb and Vd may really reflect the size control in the population. However, it is not clear why should raise them to the parameter alpha and calculate the correlation coefficient this way (page 15). What is the physiological meaning of this correlation coefficient?

14.) Are the experimental data in Figs. 5 and 6 and Table 3 from reference [11]? It should be indicated then, however, I could not find these bimodal distributions in that paper, even not in its supplementary material.

15.) The general conclusions for the model fittings seem to be quite correct concerning the sizer-like behaviour and the durations of the cell cycle phases (page 19), but some literature data should be given for comparison here. The way the authors repeat these findings in the Discussion (points (iii) and (iv)) is incorrect.

16.) Speaking about "slow growth" in the elongation phase and "rapid growth" in the fission phase (Discussion) is meaningless. See also 9.).

17.) We may say that size control seems to be stronger in EMM than in YE medium. However, speaking about strong size control in EMM and weak one in YE (Discussion) is meaningless.

18.) Although I have concerns about the model described in this paper, I admit that the simulations were perfectly fitted to the experiments, both in examining size distributions and size control parameters (Figs. 5, 6, Table 3). I can imagine a third method to study how adequate the model is. In the literature the authors may find cell length growth patterns for either representative individual fission yeast cells or for hypothetical "average" cells, measured in similar conditions. Could the authors fit their model to some cell length growth patterns? To my mind, such a presentation might really be convincing.

Reviewer #3: The paper by Jia et al. proposes a 3 stage model (and one variant of the model) for cell growth in fission yeast, provides analytical solutions and fits the model to existing data in different growth conditions. Overall, this is a in interesting paper within the remit of PLoS Comp Biology. However, the paper could be much improved after some revisions. I have the following commnets:

Major comments:

- The paper claims that the good fit of the model to data provides support for the model assumptions. However, they do not show any alternative simpler models that could fails capturing the bimodality of the data. Maybe, best fit results for one stage, two stage or alternative 3 stage models could be shown? Also, the point of the two models (model I and II) is very unclear, as the author’s do not make any attempt to rule one out. If they are both good, I suggest to move model II results, completely to Supplement and just have a paragraph on that in the main paper. I appreciate the analytical results are cool and makes inference easier, but they are not that relevant to the science.

- Instead of current Figure 6, what is more helpful is to turn some of the results in table II into graphs. What is happening to \\alpha as a function of growth rate, How is the fraction of non-growing phase changes as a function of cell cycle time? Here is some of the biologically relevant results that is now buried on a big table of numbers.

Minor comments:

- There is an old and large literature that suggests Bi-linear growth in fission yeast and the concept of NETO, more recent data has confirmed exponential growth of mass but some changes of cell density (Fred Cheng papers). This could be briefly discussed in the introductions.

- The author’s state on page 10: “when N is very large … “. An illustration would be helpful, e.g. as part of Fig 3a, N=600

- Why is the fission phase modelled as the end of the cell cycle, rather than the beginning? And why is this exponential? Atilgan ea (Curr Biol 2015) studied the new-end formation and the role of turgor pressure and might be relevant. If the size increase during fission is caused by a quick expansion of the new-end hemisphere, there is an upper bound on the added size during this phase, namely the size of a hemispherical end cap.

- The authors state on page 17: “an interesting characteristic implied by the fission yeast data ..”. This is unclear. More generally I find the distinction between model 1 and 2 unclear. Is there statistical support in the data for asymmetry? Or the data could purely be explained by stochastic partitioning.

- The proportion of elongating cells seems a little lower than estimates of the proportion of cells in G2 phase (Mitchison and Nurse, J Cell Sci 1985; Carlson ea, J Cell Sci 1999).

- The table 2, only has N values but not N_0 and N_1, why is that? Could that be added for completeness.

**Have the authors made all data and (if applicable) computational code underlying the findings in their manuscript fully available?**

Reviewer #1: **No: **I did not find where the code is available (the equations are clearly available in the main manuscript and supplement but not the underlying code).

Reviewer #2: Yes

Reviewer #3: None

PLOS authors have the option to publish the peer review history of their article (what does this mean?). If published, this will include your full peer review and any attached files.

Reviewer #1: No

Reviewer #2: No

Reviewer #3: No
---

## [Decision Letter · Decision Letter 1]

30 Nov 2021

Dear Prof. Grima,

Thank you very much for submitting your manuscript "Characterizing non-exponential growth and bimodal cell size distributions in fission yeast: an analytical approach" for consideration at PLOS Computational Biology. As with all papers reviewed by the journal, your manuscript was reviewed by members of the editorial board and by several independent reviewers. The reviewers appreciated the attention to an important topic. Based on the reviews, we are likely to accept this manuscript for publication, providing that you modify the manuscript according to the review recommendations. Please consider the changes suggested by Ref #2 and address the concern raised by Ref #1. Especially this second point requires a detailed explanation. 

Sincerely,

Attila Csikász-Nagy

Associate Editor

PLOS Computational Biology

Jason Haugh

Deputy Editor

PLOS Computational Biology

[LINK]

Reviewer's Responses to Questions

**Comments to the Authors:**

Reviewer #1: Most of my comments have been adequately addressed. However I have one major issue that has come up in the process of revision that needs to be dealt with in some way. In my initial review I requested the authors plot the data they are using as “ the relationship between birth size and ∆volume”. This was because this is the classically defined relationship that describes size homeostasis in fission yeast. This has been reported many times and always shows a clear negative correlation with a slope of -0.7 to -1. Firstly the authors should also compute and report the slope data to allow comparison with prior publications. But more crucially it does appear from this new plot (Fig. 1d) that the data the authors are using shows a significantly different relationship to all other previously published results. This suggests there may be a critical problem with the authors choice of data and I would strongly encourage them to validate their model with another dataset in which the anti-correlation between birth and growth volume is as expected.

Reviewer #2: In the revised version, the authors have made serious efforts to increase the clearness of the paper. They have either fixed or explained nearly all the mistakes or points raised by me, doubled the reference list, etc. Some small problems remained, however, which were better to be fixed before the publication of this valuable paper.

1.) In Fig. 1. the cell length were better to be shown instead of area. Based on Table 2, the cell length must heve been calculated correctly, however, the formula (area = length x width) given in the label to Fig. 1 is incorrect. The cell is a 3D cylinder (rather than a 2D rectangle).

2.) The data given in Table 2 clearly indicate (corresponding to literature data) that bimodality is a result of that the "reshaping phase" strangely belongs here to the "old" cell cycle, meanwhile generally it is supposed to belong to the next one. The calculated septation size is about 13-14 micrometer, while the division size is about 15-16 micrometer. By contrast, what experimentalists in this field generally define as division size is only 13-14 micrometer. This fact should be explicitly given in the paper.

3.) In Table 2, the generation time (T) at 34 degrees C in EMM is given as 3.633 h. That seems to be too long for me, it should probably be 2.633 h or something like that.

4.) I still cannot understand how the authors of this paper (and also of the Nakaoto-Wakamoto paper) could clearly define the timing of the end of the rounding off of the new cell poles (although they tried to do so in the revised version). No more want I to push the old hypothesis that the "reshaping phase" belongs to the new cell cycle, but I still have a feeling that the timing of the "onset" of this rounding off event is more clearly visible than that of its "finish". Please disprove my idea if possible; otherwise this problem is no more only theoretical, as it has experimental consequences as well.

5.) The above mentioned rounding off of the new cell poles is a consequence of the turgor pressure. Although sometimes it is called a hydrostatic pressure (even in textbooks, not only in this paper), I have a feeling that this is a bit incorrect.

6.) In the revised version, the authors correctly analysed the "activator accumulation" model, which may explain how size control mechanisms operate. However, there is another alternative hypothesis based in the "inhibitor dilution" model, which might also be correct, even in the case of the fission yeast cell cycle. I suggest that this should be mentioned in the paper.

Reviewer #3: The revisions are satisfactory.

**Have the authors made all data and (if applicable) computational code underlying the findings in their manuscript fully available?**

Reviewer #1: Yes

Reviewer #2: None

Reviewer #3: Yes

PLOS authors have the option to publish the peer review history of their article (what does this mean?). If published, this will include your full peer review and any attached files.

Reviewer #1: No

Reviewer #2: No

Reviewer #3: **Yes: **Vahid Shahrezaei

Figure Files:

Data Requirements:

Reproducibility:

References:

---

## [Editor Report · Decision Letter 2]

23 Dec 2021

Dear Prof. Grima,

We are pleased to inform you that your manuscript 'Characterizing non-exponential growth and bimodal cell size distributions in fission yeast: an analytical approach' has been provisionally accepted for publication in PLOS Computational Biology.

Best regards,

Attila Csikász-Nagy

Associate Editor

PLOS Computational Biology

Jason Haugh

Deputy Editor

PLOS Computational Biology

---

## [Editor Report · Acceptance letter]

13 Jan 2022

PCOMPBIOL-D-21-01114R2 

Characterizing non-exponential growth and bimodal cell size distributions in fission yeast: an analytical approach

Dear Dr Grima,

I am pleased to inform you that your manuscript has been formally accepted for publication in PLOS Computational Biology. Your manuscript is now with our production department and you will be notified of the publication date in due course.

With kind regards,

Anita Estes
